# Doubly Robust Thompson Sampling with Linear Payoffs

**Wonyoung Kim**
Department of Statistics
Seoul National University
eraser347@snu.ac.kr

**Gi-Soo Kim**
Department of Industrial Engineering &
Artificial Intelligence Graduate School
UNIST
gisookim@unist.ac.kr

**Myunghee Cho Paik**
Department of Statistics
Seoul National University
Shepherd23 Inc.
myungheechopaik@snu.ac.kr

## Abstract

A challenging aspect of the bandit problem is that a stochastic reward is observed only for the chosen arm and the rewards of other arms remain missing. The dependence of the arm choice on the past context and reward pairs compounds the complexity of regret analysis. We propose a novel multi-armed contextual bandit algorithm called Doubly Robust (DR) Thompson Sampling employing the doubly-robust estimator used in missing data literature to Thompson Sampling with contexts (LinTS). Different from previous works relying on missing data techniques (Dimakopoulou et al. [2019], Kim and Paik [2019]), the proposed algorithm is designed to allow a novel additive regret decomposition leading to an improved regret bound with the order of $\tilde{O}(\phi^{-2}\sqrt{T})$, where $\phi^2$ is the minimum eigenvalue of the covariance matrix of contexts. This is the first regret bound of LinTS using $\phi^2$ without the dimension of the context, $d$. Applying the relationship between $\phi^2$ and $d$, the regret bound of the proposed algorithm is $\tilde{O}(d\sqrt{T})$ in many practical scenarios, improving the bound of LinTS by a factor of $\sqrt{d}$. A benefit of the proposed method is that it utilizes all the context data, chosen or not chosen, thus allowing to circumvent the technical definition of unsaturated arms used in theoretical analysis of LinTS. Empirical studies show the advantage of the proposed algorithm over LinTS.

## 1 Introduction

Contextual bandit has been popular in sequential decision tasks such as news article recommendation systems. In bandit problems, the learner sequentially pulls one arm among multiple arms and receives random rewards on each round of time. While not knowing the compensation mechanisms of rewards, the learner should make his/her decision to maximize the cumulative sum of rewards. In the course of gaining information about the compensation mechanisms through feedback, the learner should carefully balance between exploitation, pulling the best arm based on information accumulated so far, and exploration, pulling the arm that will assist in future choices, although it does not seem to be the best option at the moment. Therefore in the bandit problem, estimation or learning is an important element besides decision making.

35th Conference on Neural Information Processing Systems (NeurIPS 2021).

Table 1: The shaded data are used in *complete record analysis* (left) and DR method (right) under multi-armed contextual bandit settings. The contexts, rewards and DR imputing values are denoted by $X$, $Y$, and $Y^{DR}$, respectively. The question mark refers to the missing reward of unchosen arms.

| | $t=1$ | | $t=2$ | | | $t=1$ | | $t=2$ | |
|---|---|---|---|---|---|---|---|---|---|
| Arm 1 | $X_1(1)$ | ? | $X_1(2)$ | ? | Arm 1 | $X_1(1)$ | $Y_1^{DR}(1)$ | $X_1(2)$ | $Y_1^{DR}(2)$ |
| Arm 2 | $X_2(1)$ | ? | $X_{a_2}(2)$ | $Y_{a_2}(2)$ | Arm 2 | $X_2(1)$ | $Y_2^{DR}(1)$ | $X_{a_2}(2)$ | $Y_{a_2}^{DR}(2)$ |
| Arm 3 | $X_{a_1}(1)$ | $Y_{a_1}(1)$ | $X_3(2)$ | ? | Arm 3 | $X_{a_1}(1)$ | $Y_{a_1}^{DR}(1)$ | $X_3(2)$ | $Y_3^{DR}(2)$ |
| Arm 4 | $X_4(1)$ | ? | $X_4(2)$ | ? | Arm 4 | $X_4(1)$ | $Y_4^{DR}(1)$ | $X_4(2)$ | $Y_4^{DR}(2)$ |

A challenging aspect of estimation in the bandit problem is that a stochastic reward is observed only for the chosen arm. Consequently, only the context and reward pair of the chosen arm is used for estimation, which causes dependency of the context data at the round on the past contexts and rewards. To handle this difficulty, we view bandit problems as missing data problems. The first step in handling missing data is to define full, observed, and missing data. In bandit settings, full data consist of rewards and contexts of all arms; observed data consist of full contexts for all arms and the reward for the chosen arm; missing data consist of the rewards for the arms that are not chosen. Typical estimation procedures require both rewards and contexts pairs to be observed, and the observed contexts from the unselected are discarded (see Table 1). The analysis based on the completely observed pairs only is called *complete record analysis*. Most stochastic bandit algorithms utilize estimates based on *complete record analysis*. Estimators from *complete record analysis* are known to be inefficient. In bandit setting, using the observed data whose probability of observation depends on previous rewards requires special theoretical treatment.

There are two main approaches to missing data: imputation and inverse probability weighting (IPW). Imputation is to fill in the predicted value of missing data from a specified model, and IPW is to use the observed records only but weight them by the inverse of the observation probability. The doubly robust (DR) method [Robins et al., 1994, Bang and Robins, 2005] is a combination of imputation and IPW tools. We provide a review of missing data and DR methods in supplementary materials. The robustness against model misspecification in missing data settings is insignificant in the bandit setting since the probability of observation or allocation to an arm is known. The merit of the DR method in the bandit setting is its ability to employ all the contexts including unselected arms.

We propose a novel multi-armed contextual bandit algorithm called Doubly Robust Thompson Sampling (DRTS) that applies the DR technique used in missing data literature to Thompson Sampling with linear contextual bandits (LinTS). The main thrust of DRTS is to utilize contexts information for all arms, not just chosen arms. By using the unselected, yet observed contexts, along with a novel algorithmic device, the proposed algorithm renders a unique regret decomposition which leads to a novel regret bound without resorting to the technical definition of unsaturated arms used by Agrawal and Goyal [2014]. Since categorizing the arms into saturated vs. unsaturated plays a critical role in costing extra $\sqrt{d}$, by circumventing it, we prove a $\tilde{O}(d\sqrt{T})$ bound of the cumulative regret in many practical occasions compared to $\tilde{O}(d^{3/2}\sqrt{T})$ shown in Agrawal and Goyal [2014].

The main contributions of this paper are as follows.

- We propose a novel contextual bandit algorithm that improves the cumulative regret bound of LinTS by a factor of $\sqrt{d}$ (Theorem 1) in many practical scenarios (Section 4.1). This improvement is attained mainly by defining a novel set called *super-unsaturated* arms, that is utilizable due to the proposed estimator and resampling technique adopted in the algorithm.

- We provide a novel estimation error bound of the proposed estimator (Theorem 3) which depends on the minimum eigenvalue of the covariance matrix of the contexts from all arms without $d$.

- We develop a novel dimension-free concentration inequality for sub-Gaussian vector martingale (Lemma 4) and use it in deriving our regret bound in place of the self-normalized theorem by Abbasi-Yadkori et al. [2011].

- We develop a novel concentration inequality for the bounded matrix martingale (Lemma 6) which improves the existing result (Proposition 5) by removing the dependency on $d$ in the bound. Lemma 6 also allows eliminating the forced sampling phases required in some bandit algorithms relying on Proposition 5 [Amani et al., 2019, Bastani and Bayati, 2020].

All missing proofs are in supplementary materials.

## 2 Related works

Thompson Sampling [Thompson, 1933] has been extensively studied and shown solid performances in many applications (e.g. Chapelle and Li [2011]). Agrawal and Goyal [2013] is the first to prove theoretical bounds for `LinTS` and an alternative proof is given by Abeille et al. [2017]. Both papers show $\tilde{O}(d^{3/2}\sqrt{T})$ regret bound, which is known as the best regret bound for `LinTS`. Recently, Hamidi and Bayati [2020] points out that $\tilde{O}(d^{3/2}\sqrt{T})$ could be the best possible one can get when the estimator used by `LinTS` is employed. In our work, we improve this regret bound by a factor of $\sqrt{d}$ in many practical scenarios through a novel definition of super-unsaturated arms, which becomes utilizable due to the proposed estimator and resampling device implemented in the algorithm.

Our work assumes the independence of the contexts from all arms across time rounds. Some notable works have used the assumption that the contexts are independently identically distributed (IID). Leveraging the IID assumption with a margin condition, Goldenshluger and Zeevi [2013] derives a two-armed linear contextual bandit algorithm with a regret upper bound of order $O(d^3 \log T)$. Bastani and Bayati [2020] has extended this algorithm to any number of arms and improves the regret bound to $O(d^2 \log^{\frac{3}{2}} d \cdot \log T)$. The margin condition states that the gap between the expected rewards of the optimal arm and the next best arm is nonzero with some constant probability. This condition is crucial in achieving a $O(\log T)$ regret bound instead of $\tilde{O}(\sqrt{T})$. In this paper, we do not assume this margin condition, and focus on the dependence on the dimension of contexts $d$.

From a missing data point of view, most stochastic contextual bandit algorithms use the estimator from *complete record analysis* except Dimakopoulou et al. [2019] and Kim and Paik [2019]. Dimakopoulou et al. [2019] employs an IPW estimator that is based on the selected contexts alone. Dimakopoulou et al. [2019] proves a $\tilde{O}(d\sqrt{\epsilon^{-1}T^{1+\epsilon}N})$ regret bound for their algorithm which depends on the number of arms, $N$. Kim and Paik [2019] considers the high-dimensional settings with sparsity, utilizes a DR technique, and improves the regret bound in terms of the sparse dimension instead of the actual dimension of the context, $d$. Kim and Paik [2019] is different from ours in several aspects: the mode of exploration ($\epsilon$-greedy vs. Thompson Sampling), the mode of regularization (Lasso vs. ridge regression); and the form of the estimator. A sharp distinction between the two estimators lies in that Kim and Paik [2019] aggregates contexts and rewards over the arms although they employ all the contexts. If we apply this aggregating estimator and DR-Lasso bandit algorithm to the low-dimensional setting, we obtain a regret bound of order $O(\frac{Nd}{\phi^2}\sqrt{T})$ when the contexts from the arms are independent. This bound is bigger than our bound by a factor of $d$ and $N$. It is because the aggregated form of the estimator does not permit the novel regret decomposition derived in Section 4.2. The proposed estimator coupled with a novel algorithmic device renders the additive regret decomposition which in turn improves the order of the regret bound.

## 3 Proposed estimator and algorithm

### 3.1 Settings and assumptions

We denote a $d$-dimensional context for the $i^{th}$ arm at round $t$ by $X_i(t) \in \mathbb{R}^d$, and the corresponding random reward by $Y_i(t)$ for $i = 1, \ldots, N$. We assume $\mathbb{E}\left[Y_i(t) \middle| X_i(t)\right] = X_i(t)^T \beta$ for some unknown parameter $\beta \in \mathbb{R}^d$. At round $t$, the arm that the learner chooses is denoted by $a_t \in \{1, \ldots, N\}$, and the optimal arm by $a_t^* := \arg \max_{i=1,\ldots,N} \left\{X_i(t)^T \beta\right\}$. Let $regret(t)$ be the difference between the expected reward of the chosen arm and the optimal arm at round $t$, i.e., $regret(t) := X_{a_t^*}(t)^T \beta - X_{a_t}(t)^T \beta$. The goal is to minimize the sum of regrets over $T$ rounds, $R(T) := \sum_{t=1}^T regret(t)$. The total round $T$ is finite but possibly unknown. We also make the following assumptions.

**Assumption 1. Boundedness for scale-free regrets.** For all $i = 1, \ldots, N$ and $t = 1, \ldots, T$, we have $\|X_i(t)\|_2 \leq 1$ and $\|\beta\|_2 \leq 1$.

**Assumption 2. Sub-Gaussian error.** Let $\mathcal{H}_t := \bigcup_{\tau=1}^{t-1} \left[\{X_i(\tau)\}_{i=1}^N \cup \{a_\tau\} \cup \{Y_{a_\tau}(\tau)\}\right] \cup \{X_i(t)\}_{i=1}^N$ be the set of observed data at round $t$. For each $t$ and $i$, the error $\eta_i(t) := Y_i(t) - X_i(t)^T \beta$

is conditionally zero-mean $\sigma$-sub-Gaussian for a fixed constant $\sigma \geq 0$, i.e, $\mathbb{E}\left[\eta_i(t)|\mathcal{H}_t\right] = 0$ and $\mathbb{E}\left[\exp\left(\lambda\eta_i(t)\right)|\mathcal{H}_t\right] \leq \exp(\lambda^2\sigma^2/2)$, for all $\lambda \in \mathbb{R}$. Furthermore, the distribution of $\eta_i(t)$ does not depend on the choice at round $t$, i.e. $a_t$.

**Assumption 3. Independently distributed contexts.** The stacked contexts vectors $\{X_i(1)\}_{i=1}^N, \ldots, \{X_i(T)\}_{i=1}^N \in \mathbb{R}^{dN}$ are independently distributed.

**Assumption 4. Positive minimum eigenvalue of the average of covariance matrices.** For each $t$, there exists a constant $\phi^2 > 0$ such that $\lambda_{\min}\left(\mathbb{E}\left[\frac{1}{N}\sum_{i=1}^N X_i(t)X_i(t)^T\right]\right) \geq \phi^2$.

Assumptions 1 and 2 are standard in stochastic bandit literature Agrawal and Goyal [2013]. We point out that given round $t$, Assumption 3 allows that the contexts among different arms, $X_1(t), \ldots, X_N(t)$ are correlated to each other. Assumption 3 is weaker than the assumption of IID, and the IID condition is considered by Goldenshluger and Zeevi [2013] and Bastani and Bayati [2020]. As Bastani and Bayati [2020] points out, the IID assumption is reasonable in some practical settings, including clinical trials, where health outcomes of patients are independent of those of other patients. Both Goldenshluger and Zeevi [2013] and Bastani and Bayati [2020] address the problem where the contexts are equal across all arms, i.e. $X(t) = X_1(t) = \ldots = X_N(t)$, while our work admits different contexts over all arms. Assumption 4 guarantees that the average of covariance matrices of contexts over the arms is well-behaved so that the inverse of the sample covariance matrix is bounded by the spectral norm. This assumption helps controlling the estimation error of $\beta$ in linear regression models. Similar assumptions are adopted in existing works in the bandit setting [Goldenshluger and Zeevi, 2013, Amani et al., 2019, Li et al., 2017, Bastani and Bayati, 2020].

### 3.2 Doubly robust estimator

To describe the contextual bandit DR estimator, let $\pi_i(t) := \mathbb{P}\left(a_t = i|\mathcal{H}_t\right) > 0$ be the probability of selecting arm $i$ at round $t$. We define a DR pseudo-reward as

$$Y_i^{DR}(t) = \left\{1 - \frac{\mathbb{I}\left(i = a_t\right)}{\pi_i(t)}\right\}X_i(t)^T\breve{\beta}_t + \frac{\mathbb{I}\left(i = a_t\right)}{\pi_i(t)}Y_{a_t}(t), \tag{1}$$

for some $\breve{\beta}_t$ depending on $\mathcal{H}_t$. Background of missing data methods and derivation of the DR pseudo-reward is provided in the supplementary material. Now, we propose our new estimator $\widehat{\beta}_t$ with a regularization parameter $\lambda_t$ as below:

$$\widehat{\beta}_t = \left(\sum_{\tau=1}^t \sum_{i=1}^N X_i(\tau)X_i(\tau)^T + \lambda_t I\right)^{-1}\left(\sum_{\tau=1}^t \sum_{i=1}^N X_i(\tau)Y_i^{DR}(\tau)\right). \tag{2}$$

Harnessing the pseudo-rewards defined in (1), we can make use of all contexts rather than just selected contexts. The DR estimator by Kim and Paik [2019] utilizes all contexts but has a different form from ours. While Kim and Paik [2019] uses Lasso estimator with pseudo-rewards *aggregated* over all arms, we use ridge regression estimator with pseudo-rewards in (1) which are defined *separately* for each $i = 1, \ldots, N$. This seemingly small but important difference in forms paves a way in rendering our unique regret decomposition and improving the regret bound.

### 3.3 Algorithm

In this subsection, we describe our proposed algorithm, `DRTS` which adapts DR technique to `LinTS`. The `DRTS` is presented in Algorithm 1. Distinctive features of `DRTS` compared to `LinTS` include the novel estimator and the resampling technique. At each round $t \geq 1$, the algorithm samples $\tilde{\beta}_i(t)$ from the distribution $N(\widehat{\beta}_{t-1}, v^2 V_{t-1}^{-1})$ for each $i$ independently. Let $\tilde{Y}_i(t) := X_i(t)^T\tilde{\beta}_i(t)$ and $m_t := \arg\max_i \tilde{Y}_i(t)$. We set $m_t$ as a candidate action and compute $\tilde{\pi}_{m_t}(t) := \mathbb{P}(\tilde{Y}_{m_t}(t) = \max_i \tilde{Y}_i(t)|\mathcal{H}_t)$. [1] If $\tilde{\pi}_{m_t}(t) > \gamma$, then the arm $m_t$ is selected, i.e., $a_t = m_t$. Otherwise, the algorithm resamples $\tilde{\beta}_i(t)$ until it finds another arm satisfying $\tilde{\pi}_i(t) > \gamma$ up to a predetermined fixed value $M_t$. Section A.3 in supplementary materials describes issues related to $M_t$ including a suitable choice of $M_t$.

---

[1]This computation is known to be challenging but employing the independence among $\tilde{\beta}_1(t), \ldots, \tilde{\beta}_N(t)$, we derive an explicit form approximating $\tilde{\pi}_{m_t}(t)$ in supplementary materials Section H.1.

**Algorithm 1** Doubly Robust Thompson Sampling for Linear Contextual Bandits (DRTS)

---

**Input:** Exploration parameter $v > 0$, Regularization parameter $\lambda > 0$, Selection probability threshold $\gamma \in [1/(N+1), 1/N)$, Imputation estimator $\breve{\beta}_u = f(\{X(\tau), Y_{a_\tau}(\tau)\}_{\tau=1}^{u-1})$, Number of maximum possible resampling $M_t$.

Set $F_0 = 0$, $W_0 = 0$, $\widehat{\beta}_0 = 0$ and $V_0 = \lambda I$

**for** $t = 1$ **to** $T$ **do**

    Observe contexts $\{X_i(t)\}_{i=1}^N$.

    Sample $\tilde{\beta}_1(t), \ldots, \tilde{\beta}_N(t)$ from $N(\widehat{\beta}_{t-1}, v^2 V_{t-1}^{-1})$ independently. Compute $\tilde{Y}_i(t) = X_i(t)^T \tilde{\beta}_i(t)$

    Observe a candidate action $m_t := \arg\max_i \tilde{Y}_i(t)$.

    Compute $\tilde{\pi}_{m_t}(t) := \mathbb{P}\left(\max_i \tilde{Y}_i(t) = \tilde{Y}_{m_t}(t) \middle| \mathcal{H}_t\right)$.

    **for** $l = 1$ **to** $M_t$ **do**

        **if** $\tilde{\pi}_{m_t}(t) \leq \gamma$ **then**

            Sample another $\tilde{\beta}_1(t), \ldots, \tilde{\beta}_N(t)$, observe another $m_t$, and update $\tilde{\pi}_{m_t}(t)$.

        **else**

            Break.

        **end if**

    **end for**

    Set $a_t = m_t$, and play arm $a_t$.

    Observe reward $Y_{a_t}(t)$ and compute $Y_i^{DR}(t)$

    $F_t = F_{t-1} + \sum_{i=1}^N X_i(t)Y_i^{DR}(t)$; $W_t = W_{t-1} + \sum_{i=1}^N X_i(t)X_i(t)^T$; $V_t = W_t + \lambda\sqrt{t}I$

    $\widehat{\beta}_t = V_t^{-1} F_t$

    Update $\breve{\beta}_{t+1}$ for next round.

**end for**

---

The resampling step is incorporated to avoid small values of the probability of selection so that the pseudo-reward in (1) is numerically stable. A naive remedy to stabilize the pseudo-reward is to use $\max\{\pi_i(t), \gamma\}$, which fails to leading to our regret bound since it induces bias and also cannot guarantee that the selected arm is in the super-unsaturated arms defined in (5) with high probability (For details, see Section 4.2). The resampling step implemented in the proposed algorithm is designed to solve these problems.

## 4 Theoretical results

Our theoretical results are organized as follows. In Section 4.1, we provide the main result, the cumulative regret bound of $\tilde{O}(\phi^{-2}\sqrt{T})$ of DRTS. The main thrust of deriving the regret bound is to define super-unsaturated arms. In Section 4.2 we introduce the definition of super-unsaturated arms and show how it admits a novel decomposition of the regret into two additive terms as in (6). In Section 4.3 we bound each term of the decomposed regret bounds (6). The first term is the estimation error, and Theorem 3 finds its bound. In the course of proving Theorem 3, we need Lemma 4, which plays a similar role to the self-normalized theorem of Abbasi-Yadkori et al. [2011]. We conclude the section by presenting Lemma 6 and bound the second term of (6).

### 4.1 An improved regret bound

Theorem 1 provides the regret bound of DRTS in terms of the minimum eigenvalue without $d$.

**Theorem 1.** *Suppose that Assumptions 1-4 hold. If $\breve{\beta}_t$ in Algorithm 1 satisfies $\|\breve{\beta}_t - \beta\|_2 \leq b$ for a constant $b > 0$, for all $t = 1, \ldots, T$, then with probability $1 - 2\delta$, the cumulative regret by time $T$ for DRTS algorithm is bounded by*

$$R(T) \leq 2 + \frac{4C_{b,\sigma}}{\phi^2}\sqrt{T \log \frac{12T^2}{\delta}} + \frac{2\sqrt{2T}}{\phi\sqrt{N}}, \tag{3}$$

*where $C_{b,\sigma}$ is a constant which depends only on $b$ and $\sigma$.*

The bound (3) has a rate of $O(\phi^{-2}\sqrt{T})$. The relationship between the dimension $d$ and the minimum eigenvalue $\phi^2$ can be shown by

$$d\phi^2 = \frac{d}{N}\lambda_{\min}\left(\mathbb{E}\sum_{i=1}^N X_i(t)X_i(t)^T\right) \leq \frac{1}{N}\mathbb{E}\sum_{i=1}^N \mathrm{Tr}\left(X_i(t)X_i(t)^T\right) = \frac{1}{N}\mathbb{E}\sum_{i=1}^N \|X_i(t)\|_2^2 \leq 1.$$

This implies $\phi^{-2} \geq d$, [2] but there are many practical scenarios such that $\phi^{-2} = O(d)$ holds. Bastani et al. [2021] identifies such examples including the uniform distribution and truncated multivariate normal distributions. When the context has uniform distribution on the unit ball, $\phi^{-2} = d+2$. When the context has truncated multivariate normal distribution with mean 0 and covariance $\Sigma$, we can set $\phi^{-2} = (d+2)\exp(\frac{1}{2\lambda_{\min}(\Sigma)})$. For more examples, we refer to Bastani et al. [2021]. Furthermore, regardless of distributions, $\phi^{-2} = O(d)$ holds when the correlation structure has the row sum of off-diagonals independent of the dimension, for example, AR(1), tri-diagonal, block-diagonal matrices. In these scenarios, the regret bound in (3) becomes $\tilde{O}(d\sqrt{T})$. Compared to the previous bound of LinTS [Agrawal and Goyal, 2014, Abeille et al., 2017], we obtain a better regret bound by the factor of $\sqrt{d}$ for identified practical cases.

As for the imputation estimator $\check{\beta}_t$, we assume that $\|\check{\beta}_t - \beta\|_2 \leq b$, where $b$ is an absolute constant. We suggest two cases which guarantee this assumption. First, if a biased estimator is used, we can rescale the estimator so that its $l_2$-norm is bounded by some constant $C > 0$. Then, $\|\check{\beta}_t - \beta\|_2 \leq \|\check{\beta}_t\|_2 + \|\beta\|_2 \leq C + 1$ and $b = C + 1$. Second, consistent estimators such as ridge estimator or the least squared estimator satisfy the condition since $\|\check{\beta}_t - \beta\|_2 = O(d\sqrt{\log t/t})$. The term $d$ is cancelled out when $t \geq t_d$, where $t_d$ is the minimum integer that satisfies $\log t/t \leq d^{-2}$. In these two cases, we can find a constant $b$ which satisfies the assumption on the imputation estimator $\check{\beta}_t$.

## 4.2 Super-unsaturated arms and a novel regret decomposition

The key element in deriving (3) is to decompose the regret into two additive terms as in (6). To allow such decomposition to be utilizable, we need to define a novel set of arms called super-unsaturated arms, which replaces the role of unsaturated arms in [Agrawal and Goyal, 2014]. The super-unsaturated arms are formulated so that the chosen arm is included in this set with high probability. For each $i$ and $t$, let $\Delta_i(t) := X_{a_t^*}(t)^T\beta - X_i(t)^T\beta$. Define $A_t := \sum_{\tau=1}^t X_{a_\tau}(\tau)X_{a_\tau}(\tau)^T + \lambda I$ and $V_t := \sum_{\tau=1}^t \sum_{i=1}^N X_i(\tau)X_i(\tau)^T + \lambda_t I$. For the sake of contrast, recall the definition of unsaturated arms by Agrawal and Goyal [2014]

$$U_t := \left\{i : \Delta_i(t) \leq g_t \|X_i(t)\|_{A_{t-1}^{-1}}\right\}, \tag{4}$$

where $g_t := C\sqrt{d\log(t/\delta)}\min\{\sqrt{d}, \sqrt{\log N}\}$ for some constant $C > 0$. This $g_t$ is constructed to ensure that there exists a positive lower bound for the probability that the selected arm is unsaturated. In place of (4), we define a set of super-unsaturated arms for each round $t$ by

$$N_t := \left\{i : \Delta_i(t) \leq 2\left\|\hat{\beta}_{t-1} - \beta\right\|_2 + \sqrt{\|X_{a_t^*}(t)\|_{V_{t-1}^{-1}}^2 + \|X_i(t)\|_{V_{t-1}^{-1}}^2}\right\}. \tag{5}$$

While $g_t\|X_i(t)\|_{A_{t-1}^{-1}}$ in (4) is normalized with only selected contexts, the second term in the right hand side of (5) is normalized with all contexts including $X_{a_t^*}(t)$, the contexts of the optimal arm. This bound of $\Delta_i(t)$ plays a crucial role in bounding the regret with a novel decomposition as in (6). The following Lemma shows a lower bound of the probability that the candidate arm is super-unsaturated.

**Lemma 2.** *For each $t$, let $m_t := \arg\max_i \tilde{Y}_i(t)$ and let $N_t$ be the super-unsaturated arms defined in (5). For any given $\gamma \in [1/(N+1), 1/N)$, set $v = (2\log(N/(1-\gamma N)))^{-1/2}$. Then, $\mathbb{P}(m_t \in N_t | \mathcal{H}_t) \geq 1 - \gamma$.*

Lemma 2 directly contributes to the reduction of $\sqrt{d}$ in the hyperparamter $v$. In Agrawal and Goyal [2014], to prove a lower bound of $\mathbb{P}(a_t \in U_t | \mathcal{H}_t)$, it is required to set $v = \sqrt{9d\log(t/\delta)}$, with the

---

[2]Some previous works assume $\phi^{-2} = O(1)$ even when $\|X_i(t)\|_2 \leq 1$ (e.g. Li et al. [2017]). As pointed out by Ding et al. [2021], this assumption is unrealistic and the reported regret bound should be multiplied by $O(d)$.

order of $\sqrt{d}$. In contrast, Lemma 2 shows that $v$ does not need to depend on $d$ due to the definition of super-unsaturated arms in (5). In this way, we obtain a lower bound of $\mathbb{P}\left(m_t \in N_t \mid \mathcal{H}_t\right)$ without costing extra $\sqrt{d}$.

Using the lower bound, we can show that the resampling scheme allows the algorithm to choose the super-unsaturated arms with high probability. For all $i \notin N_t$,

$$\tilde{\pi}_i(t) := \mathbb{P}\left(m_t = i \mid \mathcal{H}_t\right) \leq \mathbb{P}\left(\cup_{j \notin N_t}\{m_t = j\} \mid \mathcal{H}_t\right) = \mathbb{P}\left(m_t \notin N_t \mid \mathcal{H}_t\right) \leq \gamma,$$

where the last inequality holds due to Lemma 2. Thus, in turn, if $\tilde{\pi}_i(t) > \gamma$, then $i \in N_t$. This means that $\{i : \tilde{\pi}_i(t) > \gamma\}$ is a subset of $N_t$ and

$$\{a_t \in \{i : \tilde{\pi}_i(t) > \gamma\}\} \subset \{a_t \in N_t\}.$$

Hence, the probability of the event $\{a_t \in N_t\}$ is greater than the probability of sampling any arm which satisfies $\tilde{\pi}_i(t) > \gamma$. Therefore, with resampling, the event $\{a_t \in N_t\}$ occurs with high probability. (See supplementary materials Section A for details.)

When the algorithm chooses the arm from the super-unsaturated set, i.e., when $a_t \in N_t$ happens, (5) implies

$$\Delta_{a_t}(t) \leq 2\left\|\widehat{\beta}_{t-1} - \beta\right\|_2 + \sqrt{\left\|X_{a_t^*}(t)\right\|_{V_{t-1}^{-1}}^2 + \left\|X_{a_t}(t)\right\|_{V_{t-1}^{-1}}^2}. \tag{6}$$

By definition, $\Delta_{a_t}(t) = regret(t)$ and the regret at round $t$ can be expressed as the two additive terms, which presents a stark contrast with multiplicative decomposition of the regret in Agrawal and Goyal [2014]. In section 4.3 we show how each term can be bounded with separate rate.

### 4.3 Bounds for the cumulative regret

We first bound the leading term of (6) and introduce a novel estimation error bound free of $d$ for the contextual bandit DR estimator.

**Theorem 3.** *(A dimension-free estimation error bound for the contextual bandit DR estimator.) Suppose Assumptions 1-4 hold. For each $t = 1, \ldots, T$, let $\breve{\beta}_t$ be any $\mathcal{H}_t$-measurable estimator satisfying $\|\breve{\beta}_t - \beta\|_2 \leq b$, for some constant $b > 0$. For each $i$ and $t$, assume that $\pi_i(t) > 0$ and that there exists $\gamma \in [1/(N+1), 1/N)$ such that $\pi_{a_t}(t) > \gamma$. Given any $\delta \in (0, 1)$, set $\lambda_t = 4\sqrt{2}N\sqrt{t \log \frac{12\tau^2}{\delta}}$. Then with probability at least $1 - \delta$, the estimator $\widehat{\beta}_t$ in (2) satisfies*

$$\left\|\widehat{\beta}_t - \beta\right\|_2 \leq \frac{C_{b,\sigma}}{\phi^2 \sqrt{t}} \sqrt{\log \frac{12t^2}{\delta}}, \tag{7}$$

*for all $t = 1, \ldots, T$, where the constant $C_{b,\sigma}$ which depends only on $b$ and $\sigma$.*

In bandit literature, estimation error bounds typically include a term involving $d$ which emerges from using the following two Lemmas: (i) the self-normalized bound for vector-valued martingales [Abbasi-Yadkori et al., 2011, Theorem 1], and (ii) the concentration inequality for the covariance matrix [Tropp, 2015, Corollary 5.2]. Instead of using (i) and (ii), we develop the two dimension-free bounds in Lemmas 4 and 6, to replace (i) and (ii), respectively. With the two Lemmas, we eliminate the dependence on $d$ and express the estimation error bound with $\phi^2$ alone.

**Lemma 4.** *(A dimension-free bound for vector-valued martingales.) Let $\{\mathcal{F}_\tau\}_{\tau=1}^t$ be a filtration and $\{\eta(\tau)\}_{\tau=1}^t$ be a real-valued stochastic process such that $\eta(\tau)$ is $\mathcal{F}_\tau$-measurable. Let $\{X(\tau)\}_{\tau=1}^t$ be an $\mathbb{R}^d$-valued stochastic process where $X(\tau)$ is $\mathcal{F}_{\tau-1}$-measurable and $\|X(\tau)\|_2 \leq 1$. Assume that $\{\eta(\tau)\}_{\tau=1}^t$ are $\sigma$-sub-Gaussian as in Assumption 2. Then with probability at least $1 - \delta/t^2$, there exists an absolute constant $C > 0$ such that*

$$\left\|\sum_{\tau=1}^t \eta(\tau)X(\tau)\right\|_2 \leq C\sigma\sqrt{t}\sqrt{\log \frac{4t^2}{\delta}}. \tag{8}$$

Compared to Theorem 1 of Abbasi-Yadkori et al. [2011], our bound (8) does not involve $d$, yielding a dimension-free bound for vector-valued martingales. However, the bound (8) has $\sqrt{t}$ term which comes from using $\|\cdot\|_2$ instead of the self-normalized norm $\|\cdot\|_{V_t^{-1}}$.

To complete the proof of Theorem 3, we need the following condition,

$$\lambda_{\min}\left(V_t\right) \geq ct, \tag{9}$$

for some constant $c > 0$. Li et al. [2017] points out that satisfying (9) is challenging. To overcome this difficulty, Amani et al. [2019] and Bastani and Bayati [2020] use an assumption on the covariance matrix of contexts and a concentration inequality for matrix to prove (9), described as follows.

**Proposition 5.** *[Tropp, 2015, Theorem 5.1.1] Let $P(1), \ldots, P(t) \in \mathbb{R}^{d \times d}$ be the symmetric matrices such that $\lambda_{\min}(P(\tau)) \geq 0$, $\lambda_{\max}(P(\tau)) \leq L$ and $\lambda_{\min}(\mathbb{E}[P(\tau)]) \geq \phi^2$, for all $\tau = 1, 2, \ldots, t$. Then,*

$$\mathbb{P}\left(\lambda_{\min}\left(\sum_{\tau=1}^{t} P(\tau)\right) \leq \frac{t\phi^2}{2}\right) \leq d \exp\left(-\frac{t\phi^2}{8L}\right). \tag{10}$$

To prove (9) using (10) with probability at least $1 - \delta$, for $\delta \in (0, 1)$, it requires $t \geq \frac{8L}{\phi^2} \log \frac{d}{\delta}$. Thus, one can use (10) only after $O(\phi^{-2} \log d)$ rounds. Due to this requirement, Bastani and Bayati [2020] implements the forced sampling techniques for $O\left(N^2 d^4 (\log d)^2\right)$ rounds, and Amani et al. [2019] forces to select arms randomly for $O\left(\phi^{-2} \log d\right)$ rounds. These mandatory exploration phase empirically prevents the algorithm choosing the optimal arm. An alternative form of matrix Chernoff inequality for adapted sequences is Theorem 3 in Tropp [2011], but the bound also has a multiplicative factor of $d$. Instead of applying Proposition 5 to prove (9), we utilize a novel dimension-free concentration inequality stated in the following Lemma.

**Lemma 6.** *(A dimension-free concentration bound for symmetric bounded matrices.) Let $\|A\|_F$ be a Frobenious norm of a matrix $A$. Let $\{P(\tau)\}_{\tau=1}^{t} \in \mathbb{R}^{d \times d}$ be the symmetric matrices adapted to a filtration $\{\mathcal{F}_\tau\}_{\tau=1}^{t}$. For each $\tau = 1, \ldots, t$, suppose that $\|P(\tau)\|_F \leq c$, for some $c > 0$ and $\lambda_{\min}\left(\mathbb{E}\left[P(\tau)|\mathcal{F}_{\tau-1}\right]\right) \geq \phi^2 > 0$, almost surely. For given any $\delta \in (0, 1)$, set $\lambda_t \geq 4\sqrt{2}c\sqrt{t}\sqrt{\log \frac{4t^2}{\delta}}$. Then with probability at least $1 - \delta/t^2$,*

$$\lambda_{\min}\left(\sum_{\tau=1}^{t} P(\tau) + \lambda_t I\right) \geq \phi^2 t. \tag{11}$$

Lemma 6 shows that setting $\lambda_t$ with $\sqrt{t}$ rate guarantees (9) for all $t \geq 1$. We incorporate $\lambda_t$ stated in Lemma 6 in our estimator (2), and show in Section 5 that the DR estimator regularized with $\lambda_t$ outperforms estimators from other contextual bandit algorithms in early rounds.

We obtain the bounds free of $d$ in Lemmas 4 and 6 mainly by applying Lemma 2.3 in Lee et al. [2016] which states that any Hilbert space martingale can be reduced to $\mathbb{R}^2$. Thus, we can project the vector-valued (or the matrix) martingales to $\mathbb{R}^2$-martingales, and reduce the dimension from $d$ (or $d^2$) to 2. Then we apply Azuma-Hoeffding inequality just twice, instead of $d$ times. In this way, Lemma 6 provides a novel dimension-free bound for the covariance matrix.

Lemmas 4 and 6 can be applied to other works to improve the existing bounds. For example, using these Lemmas, the estimation error bound of Bastani and Bayati [2020] can be improved by a factor of $\log d$. Proposition EC.1 of Bastani and Bayati [2020] provides an estimation error bound for the ordinary least square estimator by using Proposition 5 and bounding all values of $d$ coordinates. By applying Lemmas 4 and 6, one does not have to deal with each coordinate and eliminate dependence on $d$.

Using Lemma 6, we can bound the second term of the regret in (6) as follows. For $j = 1, \ldots, N$

$$\|X_j(t)\|_{V_{t-1}^{-1}} \leq \|X_j(t)\|_2 \sqrt{\left\|V_{t-1}^{-1}\right\|_2} \leq \lambda_{\min}\left(V_{t-1}\right)^{-1/2} \leq \frac{1}{\sqrt{\phi^2 N(t-1)}}. \tag{12}$$

Finally, we are ready to bound $regret(t)$ in (6).

**Lemma 7.** *Suppose the assumptions in Theorem 1 hold. Then with probability at least $1 - 2\delta$,*

$$regret(t) \leq \frac{2C_{b,\sigma}}{\phi^2\sqrt{t-1}}\sqrt{\log \frac{12t^2}{\delta}} + \frac{\sqrt{2}}{\phi\sqrt{N(t-1)}}, \tag{13}$$

*for all $t = 2, \ldots, T$.*

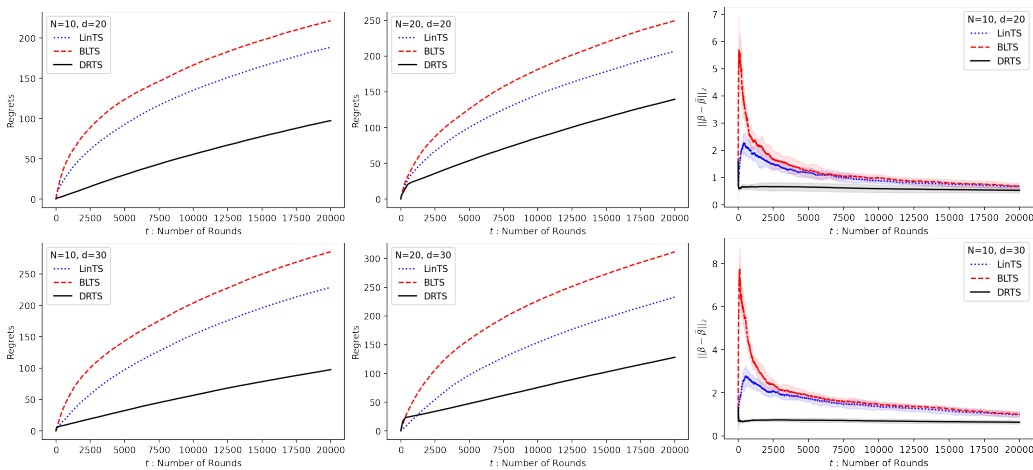

Figure 1: A Comparison of cumulative regrets and estimation errors of `LinTS`, `BLTS` and `DRTS`. Each line shows the averaged cumulative regrets (estimation errors, resp.) and the shaded area in the right two figures represents the standard deviations over 10 repeated experiments.

*Proof.* Since $a_t$ is shown to be super-unsaturated with high probability, we can use (6) to have $regret(t) \leq 2\|\widehat{\beta}_{t-1} - \beta\|_2 + \sqrt{\|X_{a_t^*}(t)\|_{V_{t-1}^{-1}}^2 + \|X_{a_t}(t)\|_{V_{t-1}^{-1}}^2}$, for all $t = 2, \ldots, T$. We see that the first term is bounded by Theorem 3, and the second term by (12). Note that to prove Theorem 1, Lemma 6 is invoked, and the event (11) of Lemma 6 is a subset of that in (7). Therefore (13) holds with probability at least $1 - 2\delta$ instead of $1 - 3\delta$. Details are given in supplementary materials. □

Lemma 7 shows that the regret at round $t$ does not exceed a $O(\phi^{-2}t^{-1/2})$ bound when $a_t \in N_t$, which is guaranteed in our algorithm via resampling with high probability (See Section A.3 for details). This concludes the proof of Theorem 1.

## 5 Simulation studies

In this section, we compare the performances of the three algorithms: (i) `LinTS` [Agrawal and Goyal, 2013], (ii) `BLTS` [Dimakopoulou et al., 2019], and (iii) the proposed `DRTS`. We use simulated data described as follows. The number of arms $N$ is set to 10 or 20, and the dimension of contexts $d$ is set to 20 or 30. For each element of the contexts $j = 1, \cdots, d$, we generate $[X_{1j}(t), \cdots, X_{Nj}(t)]$ from a normal distribution $\mathcal{N}(\mu_N, V_N)$ with mean $\mu_{10} = [-10, -8, \cdots, -2, 2, \cdots, 8, 10]^T$, or $\mu_{20} = [-20, -18, \cdots, -2, 2, \cdots, 18, 20]^T$, and the covariance matrix $V_N \in \mathbb{R}^{N \times N}$ has $V_N(i, i) = 1$ for every $i$ and $V_N(i, k) = \rho$ for every $i \neq k$. We set $\rho = 0.5$ and truncate the sampled contexts to satisfy $\|X_i(t)\|_2 \leq 1$. To generate the stochastic rewards, we sample $\eta_i(t)$ independently from $\mathcal{N}(0, 1)$. Each element of $\beta$ follows a uniform distribution, $\mathcal{U}(-1/\sqrt{d}, 1/\sqrt{d})$.

All three algorithms have $v$ as an input parameter which controls the variance of $\tilde{\beta}_i(t)$. `BLTS` and `DRTS` require a positive threshold $\gamma$ which truncates the selection probability. We consider $v \in \{0.001, 0.01, 0.1, 1\}$ in all three algorithms, $\gamma \in \{0.01, 0.05, 0.1\}$ for `BLTS`, and set $\gamma = 1/(N + 1)$ in `DRTS`. Then we report the minimum regrets among all combinations. The regularization parameter is $\lambda_t = \sqrt{t}$ in `DRTS` and $\lambda_t = 1$ in both `LinTS` and `BLTS`. To obtain an imputation estimator $\check{\beta}_t$ required in `DRTS`, we use ridge regression with $\{X_{a_\tau}(\tau), Y_{a_\tau}(\tau)\}_{\tau=1}^{t-1}$, for each round $t$. Other implementation details are in supplementary materials.

Figure 1 shows the average of the cumulative regrets and the estimation error $\|\widehat{\beta}_t - \beta\|_2$ of the three algorithms based on 10 replications. The figures in the two left columns show the average cumulative regret according to the number of rounds with the best set of hyperparameters for each algorithm. The total rounds are $T = 20000$. The figures in the third columns show the average of the estimation error $\|\widehat{\beta}_t - \beta\|_2$. In the early stage, the estimation errors of `LinTS` and `BLTS` increase rapidly, while that of `DRTS` is stable. The stability of the DR estimator follows possibly by using full contexts and

the regularization parameter $\lambda_t = \sqrt{t}$. This yields a large margin of estimation error among `LinTS`, `BLTS` and `DRTS`, especially when the dimension is large.

## 6    Conclusion

In this paper, we propose a novel algorithm for stochastic contextual linear bandits. Viewing the bandit problem as a missing data problem, we use the DR technique to employ all contexts including those that are not chosen. With the definition of super-unsaturated arms, we show a regret bound which only depends on the minimum eigenvalue of the sample covariance matrices. This new bound has $\tilde{O}(d\sqrt{T})$ rate in many practical scenarios, which is improved by a factor of $\sqrt{d}$ compared to the previous `LinTS` regret bounds. Simulation studies show that the proposed algorithm performs better than other `LinTS` algorithms in a large dimension.

## Acknowledgements

This work is supported by the National Research Foundation of Korea (NRF) grant funded by the Korea government (MSIT, No.2020R1A2C1A01011950) (Wonyoung Kim and Myunghee Cho Paik), and by the Institute of Information & communications Technology Planning & Evaluation (IITP) grant funded by the Korea government (MSIT) (No.2020-0-01336, Artificial Intelligence Graduate School Program(UNIST)) and the National Research Foundation of Korea (NRF) grant funded by the Korea government (MSIT, No.2021R1G1A100980111) (Gi-Soo Kim). Wonyoung Kim was also supported by Hyundai Chung Mong-koo foundation.

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
