# A Detailed analysis of the resampling

In this section, we give details about the issues which can be raised from the resampling in our algorithm.

## A.1 Precise definition of action selection

We give precise definition of the action at round $t$, $a_t$. For each round $t \geq 2$, given $\mathcal{H}_t$, let $a_t^{(1)}, a_t^{(2)}, \ldots, a_t^{(M_t)}$ to be maximum possible sequence of actions to be resampled. These actions are IID, with $\mathbb{P}\left(a_t^{(1)} = i \Big| \mathcal{H}_t\right) = \tilde{\pi}_i(t)$ for $i = 1, \ldots, N$. Define a subset of arms $\tilde{\Gamma}_t := \{i : \tilde{\pi}_i(t) > \gamma\}$ and a stopping time

$$\mathcal{T} := \inf\{m \geq 1 : a_t^{(m)} \in \tilde{\Gamma}_t\} \tag{14}$$

with respect to the filtration $\mathcal{F}_m := \mathcal{H}_t \cup \{a_t^{(1)}, \ldots, a_t^{(m)}\}$. Since the algorithm stops resampling when the candidate action is in $\tilde{\Gamma}_t$, the stopping time $\mathcal{T}$ is the actual number of resampling in algorithm. Thus we can write the action after resampling as $a_t := a_t^{(\min\{\mathcal{T}, M_t\})}$.

## A.2 Computing the probability of selection

The probability of selection $\pi_i(t) := \mathbb{P}\left(a_t = i \mid \mathcal{H}_t\right)$ is not the same as $\tilde{\pi}_i(t)$ due to resampling. This might cause the problem of computing $\pi_i(t)$ which is essential to compute $Y_i^{DR}(t)$. However, with the precise definition of $a_t$, we can derive a closed form for $\pi_i(t)$.

First, we consider two cases separately: (i) the case when the resampling succeeds and (ii) the case when the resampling fails and the maximum possible number of resampling runs out. In case (i), $a_t \in \tilde{\Gamma}_t$, and for any $i \in \tilde{\Gamma}_t$, we have

$$
\begin{aligned}
\mathbb{P}\left(a_t = i \mid \mathcal{H}_t\right) &= \mathbb{P}\left(\mathcal{T} \leq M_t, \ a_t^{(\mathcal{T})} = i \Big| \mathcal{H}_t\right) \\
&= \sum_{m=1}^{M_t} \mathbb{P}\left(\mathcal{T} = m, \ a_t^{(m)} = i \Big| \mathcal{H}_t\right) \\
&= \sum_{m=1}^{M_t} \mathbb{P}\left(a_t^{(m)} = i \Big| \mathcal{H}_t\right) \left(\prod_{j=0}^{m-1} \mathbb{P}\left(a_t^{(j)} \notin \tilde{\Gamma}_t \Big| \mathcal{H}_t\right)\right) \\
&= \tilde{\pi}_i(t) \sum_{m=1}^{M_t} \left(1 - \sum_{i \in \tilde{\Gamma}_t} \tilde{\pi}_i(t)\right)^{m-1} \\
&= \tilde{\pi}_i(t) \frac{1 - \left(1 - \sum_{i \in \tilde{\Gamma}_t} \tilde{\pi}_i(t)\right)^{M_t}}{\sum_{i \in \tilde{\Gamma}_t} \tilde{\pi}_i(t)}.
\end{aligned}
\tag{15}
$$

Now, for the case (ii) $a_t \notin \tilde{\Gamma}_t$, and for any $i \notin \tilde{\Gamma}_t$, we have

$$
\begin{aligned}
\mathbb{P}\left(a_t = i \mid \mathcal{H}_t\right) &= \mathbb{P}\left(\mathcal{T} > M_t, a_t^{(M_t)} = i \Big| \mathcal{H}_t\right) \\
&= \mathbb{P}\left(\bigcap_{m=1}^{M_t-1} \left\{a_t^{(m)} \notin \tilde{\Gamma}_t\right\}, a_t^{(M_t)} = i \Big| \mathcal{H}_t\right) \\
&= \left(1 - \sum_{i \in \tilde{\Gamma}_t} \tilde{\pi}_i(t)\right)^{M_t-1} \tilde{\pi}_i(t).
\end{aligned}
\tag{16}
$$

With (15) and (16), we can compute $\pi_i(t)$ for all $i = 1, \ldots, N$.

## A.3 The number of maximum possible resampling

The proposed algorithm attempts resampling up to $M_t$ times to find an arm in $\{i : \tilde{\pi}_i(t) > \gamma\}$. The main point in selecting $M_t$ is to bound the probability that the resampling fails in finding an arm whose selection probability exceeds $\gamma$ for some $\delta$, i.e.,

$$\mathbb{P}(a_t \notin \{i : \tilde{\pi}_i(t) > \gamma\}) \leq \delta/t^2. \tag{17}$$

Intuitively, as $M_t$ increases, we have more opportunities for resampling and the probability that the resampling fails in finding arms in $\{i : \tilde{\pi}_i(t) > \gamma\}$ decreases. Since $\gamma < 1/N$, there exists $j$ such that $\tilde{\pi}_j(t) > \gamma$, and the probability that the resampling fails is less than $1 - \gamma$ in each resampling trial.

Specifically, we can achieve (17) by choosing $M_t$ as a minimum integer that exceeds $\log \frac{t^2}{\delta} / \log \frac{1}{1-\gamma}$. For any given $\delta \in (0, 1)$, the event $\{a_t \in \tilde{\Gamma}_t\}$ occurs with probability at least $1 - \delta/t^2$. By (14), we have

$$\mathbb{P}\left(a_t \notin \tilde{\Gamma}_t \,\middle|\, \mathcal{H}_t\right) = \mathbb{P}\left(\mathcal{T} > M_t \,\middle|\, \mathcal{H}_t\right) = \mathbb{P}\left(\bigcap_{m=1}^{M_t} \left\{a_t^{(m)} \notin \tilde{\Gamma}_t\right\} \,\middle|\, \mathcal{H}_t\right) = \left(1 - \sum_{i \in \tilde{\Gamma}_t} \tilde{\pi}_i(t)\right)^{M_t}.$$

Since $\gamma < 1/N$, there exists at least one arm in $\tilde{\Gamma}_t$, and thus $\mathbb{P}\left(a_t \notin \tilde{\Gamma}_t \,\middle|\, \mathcal{H}_t\right) \leq (1 - \gamma)^{M_t}$. If we set $M_t$ as a minimum integer that exceeds $\left(\log \frac{t^2}{\delta}\right)\left(\log \frac{1}{1-\gamma}\right)^{-1}$ then (17) holds. Thus, by choosing $M_t$ for each round that satisfies (17), the algorithm finds an arm $j$ such that $\tilde{\pi}_j(t) > \gamma$ in all rounds with high probability.

Selecting an arm from the set $\{i : \tilde{\pi}_i(t) > \gamma\}$ with high probability is crucial in achieving the regret bound of order $\tilde{O}(\phi^{-2}\sqrt{T})$ for two reasons. First, it guarantees that the arm is super-unsaturated and our novel regret decomposition (6) holds to achieve our regret bound. Let $N_t$ be the set of super-unsaturated arm defined in (5). With Lemma 2, we prove that if $\tilde{\pi}_i(t) > \gamma$ then $i \in N_t$, which implies $\tilde{\Gamma}_t \subseteq N_t$, and thus

$$\mathbb{P}\left(a_t \in N_t \,\middle|\, \mathcal{H}_t\right) \geq \mathbb{P}\left(a_t \in \tilde{\Gamma}_t \,\middle|\, \mathcal{H}_t\right).$$

Thus we can conclude that $a_t$ is super-unsaturated with probability at least $1 - \delta/t^2$ with $M_t$ defined in Section A.3. Second, the inverse probability, $\pi_{a_t}(t)^{-1}$ is bounded by $\gamma^{-1}$ which appears in $Y_i^{DR}(t)$ and the proof of Theorem 3. From (15) we can deduce $\pi_{a_t}(t) \geq \tilde{\pi}_{a_t}(t) > \gamma$, for $a_t \in \tilde{\Gamma}_t$. This shows that the assumptions regarding $\pi_{a_t}(t)$ in Theorem 3 hold.

## B Technical Lemmas

**Lemma 8.** *[Wainwright, 2019, Theorem 2.19] (Bernstein Concentration) Let $\{D_k, \mathfrak{S}_k\}_{k=1}^{\infty}$ be a martingale difference sequence and suppose $D_k$ is $\sigma$-sub-Gaussian in an adapted sense, i.e. for all $\lambda \in \mathbb{R}$, $\mathbb{E}\left[e^{\lambda D_k} \,\middle|\, \mathfrak{S}_{k-1}\right] \leq e^{\lambda^2 \sigma^2/2}$ almost surely. Then for all $x \geq 0$,*

$$\mathbb{P}\left(\left|\sum_{k=1}^{n} D_k\right| \geq x\right) \leq 2 \exp\left(-\frac{x^2}{2n\sigma^2}\right).$$

**Lemma 9.** *[Azuma, 1967] (Azuma-Hoeffding inequality) If a super-martingale $(Y_t; t \geq 0)$ corresponding to filtration $\mathcal{F}_t$, satisfies $|Y_t - Y_{t-1}| \leq c_t$ for some constant $c_t$, for all $t = 1, \ldots, T$, then for any $a \geq 0$,*

$$\mathbb{P}\left(Y_T - Y_0 \geq a\right) \leq e^{-\frac{a^2}{2\sum_{t=1}^{T} c_t^2}}.$$

**Lemma 10.** *[Lee et al., 2016, Lemma 2.3] Let $\{N_t\}$ be a martingale on a Hilbert space $(\mathcal{H}, \|\cdot\|_{\mathcal{H}})$. Then there exists a $\mathbb{R}^2$-valued martingale $\{P_t\}$ such that for any time $t \geq 0$, $\|P_t\|_2 = \|N_t\|_{\mathcal{H}}$ and $\|P_{t+1} - P_t\|_2 = \|N_{t+1} - N_t\|_{\mathcal{H}}$.*

**Lemma 11.** *[Chung and Lu, 2006, Lemma 1, Theorem 32] For a filtration $\mathcal{F}_0 \subset \mathcal{F}_1 \subset \cdots \subset \mathcal{F}_T$, suppose each random variable $X_t$ is $\mathcal{F}_t$-measurable martingale, for $0 \leq t \leq T$. Let $B_t$ denote the bad set associated with the following admissible condition:*

$$|X_t - X_{t-1}| \leq c_t,$$

*for $1 \leq t \leq T$, where $c_1, \ldots, c_n$ are non-negative numbers. Then there exists a collection of random variables $Y_0, \ldots, Y_T$ such that $Y_t$ is $\mathcal{F}_t$-measurable martingale such that*

$$|Y_t - Y_{t-1}| \leq c_t,$$

*and $\{\omega : Y_t(\omega) \neq X_t(\omega)\} \subset B_t$, for $0 \leq t \leq T$.*

**Lemma 12.** *Suppose a random variable $X$ satisfies $\mathbb{E}[X] = 0$, and let $\eta$ be an $\sigma$-sub-Gaussian random variable. If $|X| \leq |\eta|$ almost surely, then $X$ is $C\sigma$-sub-Gaussian for some absolute constant $C > 0$.*

*Proof.* By Proposition 2.5.2 in Vershynin [2018], there exists an absolute constant $C_1 > 0$ such that

$$\mathbb{E}\exp\left(\lambda^2 \eta^2\right) \leq \exp\left(\frac{\lambda^2 C_1^2 \sigma^2}{2}\right), \quad \forall \lambda \in \left[-\frac{\sqrt{2}}{C_1 \sigma}, \frac{\sqrt{2}}{C_1 \sigma}\right].$$

Since $|X| \leq |\eta|$ almost surely,

$$\mathbb{E}\exp\left(\lambda^2 X^2\right) \leq \exp\left(\frac{\lambda^2 C_1^2 \sigma^2}{2}\right), \quad \forall \lambda \in \left[-\frac{\sqrt{2}}{C_1 \sigma}, \frac{\sqrt{2}}{C_1 \sigma}\right].$$

Since $\mathbb{E}[X] = 0$, by Proposition 2.5.2 in Vershynin [2018], there exists an absolute constant $C_2 > 0$ such that

$$\mathbb{E}\exp\left(\lambda X\right) \leq \exp\left(\frac{\lambda^2 C_1^2 C_2^2 \sigma^2}{2}\right), \quad \forall \lambda \in \mathbb{R}.$$

Setting $C = C_1 C_2$ completes the proof. $\square$

# C   Missing details in proof of Theorem 1

In section A.3, we prove that $a_t \in \tilde{\Gamma}_t$ with probability at least $1 - \delta/t^2$, for all $t \geq 2$. Thus, for any $x > 0$,

$$\mathbb{P}\left(R(T) > x\right) \leq \mathbb{P}\left(R(T) > x, \bigcap_{t=2}^{T}\left\{a_t \in \tilde{\Gamma}_t\right\}\right) + \mathbb{P}\left(\bigcup_{t=2}^{T}\left\{a_t \notin \tilde{\Gamma}_t\right\}\right)$$

$$\leq \mathbb{P}\left(R(T) > x, \bigcap_{t=2}^{T}\left\{a_t \in \tilde{\Gamma}_t\right\}\right) + \delta$$

$$\leq \mathbb{P}\left(2 + \sum_{t=2}^{T} regret(t) > x, \bigcap_{t=2}^{T}\left\{a_t \in \tilde{\Gamma}_t\right\}\right) + \delta$$

The last inequality holds by Assumption 1. Since $\tilde{\Gamma}_t$ is a subset of $N_t$ and by (6),

$$\mathbb{P}\left(R(T) > x\right)$$
$$\leq \mathbb{P}\left(2 + \sum_{t=2}^{T}\left\{2\left\|\widehat{\beta}_{t-1} - \beta\right\|_2 + \sqrt{\left\|X_{a_t^*}(t)\right\|_{V_{t-1}^{-1}}^2 + \left\|X_{a_t}(t)\right\|_{V_{t-1}^{-1}}^2}\right\} > x, \bigcap_{t=2}^{T}\left\{a_t \in \tilde{\Gamma}_t\right\}\right) + \delta.$$
$$\tag{18}$$

To bound the term $\left\|\widehat{\beta}_t - \beta\right\|_2$ for all $t = 1, \ldots, T-1$, we use Theorem 3. Before that, we need to verify whether the two assumptions on $\pi_i(t)$ in Theorem 3 hold.

First, we show that $\pi_{a_t}(t) > \gamma$. When $t = 1$, we have $\tilde{\pi}_i(1) = 1/N$ for all $i$. Since $\gamma < 1/N$, we do not need resampling and thus $\pi_i(t) = \tilde{\pi}_i(t) > \gamma$. When $t \geq 2$, $a_t \in \tilde{\Gamma}_t$ is already concerned in (18), and thus $\tilde{\pi}_{a_t}(t) > \gamma$. From (15), we can deduce that $\pi_i(t) > \tilde{\pi}_i(t)$ for all $i \in \tilde{\Gamma}_t$, and thus $\pi_{a_t}(t) > \gamma$.

Now, we prove that $\pi_i(t) > 0$ for all $i$ and $t$. The case of $t = 1$ is already proved above. When $t \geq 2$, from (15), we have

$$\pi_i(t) := \mathbb{P}\left( a_t = i \mid \mathcal{H}_t \right) = \tilde{\pi}_i(t) \sum_{m=1}^{M_t} \left( 1 - \sum_{i \in \tilde{\Gamma}_t} \tilde{\pi}_i(t) \right)^{m-1} > \tilde{\pi}_i(t) > \gamma,$$

for all $i \in \tilde{\Gamma}_t$. If there exists an arm $i \notin \tilde{\Gamma}_t$, from (16),

$$\pi_i(t) = \left( 1 - \sum_{i \in \tilde{\Gamma}_t} \tilde{\pi}_i(t) \right)^{M_t - 1} \tilde{\pi}_i(t).$$

The first term is positive since there exists an arm $i \notin \tilde{\Gamma}_t$. The second term is also positive since the distribution of $\tilde{\beta}_i(t)$ has support $\mathbb{R}^d$, which implies that

$$\tilde{\pi}_i(t) := \mathbb{P}\left( X_i(t)^T \tilde{\beta}_i(t) = \max_j X_j(t)^T \tilde{\beta}_j(t) \,\Big|\, \mathcal{H}_t \right) > 0,$$

for all $i$. Thus, $\pi_i(t) > 0$ for all $i$ and $t$. This implies that the two assumptions on $\pi_i(t)$ in Theorem 3 hold.

Now we can use Theorem 3 and Lemma 6 to have

$$\left\| \widehat{\beta}_{t-1} - \beta \right\|_2 \leq \frac{C_{b,\sigma}}{\phi^2 \sqrt{t-1}} \sqrt{\log \frac{12(t-1)^2}{\delta}}, \quad \sqrt{\left\| X_{a_t^*}(t) \right\|_{V_{t-1}^{-1}}^2 + \left\| X_{a_t}(t) \right\|_{V_{t-1}^{-1}}^2} \leq \frac{1}{\phi \sqrt{N(t-1)}},$$

for all $t = 2, \ldots, T$ with probability at least $1 - \delta$. Thus, setting

$$x = 2 + \frac{4 C_{b,\sigma}}{\phi^2} \sqrt{T \log \frac{12 T^2}{\delta}} + \frac{2 \sqrt{T}}{\phi \sqrt{N}}$$

in (18) proves the result.

## D  Proof of Lemma 2

*Proof.*  First, we bring attention to the fact that the optimal arm $a_t^*$ is in $N_t$ by definition. Suppose that the estimated reward of the optimal arm, $\tilde{Y}_{a_t^*}(t)$ is greater than $\tilde{Y}_j(t)$ for all $j \notin N_t$. In this case, any arm $j \notin N_t$ cannot be the $m_t := \arg\max_i \tilde{Y}_i(t)$. Then we have

$$\mathbb{P}\left( m_t \in N_t \mid \mathcal{H}_t \right) \geq \mathbb{P}\left( \tilde{Y}_{a_t^*}(t) > \tilde{Y}_j(t), \forall j \notin N_t \,\Big|\, \mathcal{H}_t \right)$$

$$= \mathbb{P}\left( Z_j(t) > \{ X_j(t) - X_{a_t^*}(t) \}^T \widehat{\beta}_{t-1}, \forall j \notin N_t \,\Big|\, \mathcal{H}_t \right),$$

where $Z_j(t) := \tilde{Y}_{a_t^*}(t) - \tilde{Y}_j(t) - \{ X_{a_t^*}(t) - X_j(t) \}^T \widehat{\beta}_{t-1}$. Note that $Z_j(t)$ is a Gaussian random variable with mean 0 and variance $v^2 (\| X_{a_t^*}(t) \|_{V_{t-1}^{-1}}^2 + \| X_j(t) \|_{V_{t-1}^{-1}}^2)$ given $\mathcal{H}_t$. For all $j \notin N_t$,

$$\{ X_j(t) - X_{a_t^*}(t) \}^T \widehat{\beta}_{t-1} = \{ X_j(t) - X_{a_t^*}(t) \}^T \{ \widehat{\beta}_{t-1} - \beta \} - \Delta_j(t)$$

$$\leq 2 \left\| \widehat{\beta}_t - \beta \right\|_2 - \Delta_j(t) \leq -\sqrt{\left\| X_{a_t^*}(t) \right\|_{V_{t-1}^{-1}}^2 + \left\| X_j(t) \right\|_{V_{t-1}^{-1}}^2}.$$

The last inequality is due to $j \notin N_t$. Thus, we can conclude that

$$\mathbb{P}\left( m_t \in N_t \mid \mathcal{H}_t \right) \geq \mathbb{P}\left( \frac{Z_j(t)}{v \sqrt{\left\| X_{a_t^*}(t) \right\|_{V_{t-1}^{-1}}^2 + \left\| X_j(t) \right\|_{V_{t-1}^{-1}}^2}} > -\frac{1}{v}, \forall j \notin N_t \,\Bigg|\, \mathcal{H}_t \right)$$

$$:= \mathbb{P}\left( Y_j > -v^{-1}, \forall j \neq N_t \mid \mathcal{H}_t \right).$$

Using the fact that

$$Y_j := \frac{Z_j(t)}{v\sqrt{\left\|X_{a_t^*}(t)\right\|_{V_{t-1}^{-1}}^2 + \left\|X_j(t)\right\|_{V_{t-1}^{-1}}^2}}$$

is a standard Gaussian random variable given $\mathcal{H}_t$, we have

$$\mathbb{P}\left(Y_j \leq -v^{-1}\middle|\mathcal{H}_t\right) \leq \exp\left(-\frac{1}{2v^2}\right).$$

Setting $v = \{2\log(N/(1-\gamma N))\}^{-1/2}$ gives

$$\mathbb{P}\left(Y_j \leq -v^{-1}\middle|\mathcal{H}_t\right) \leq \exp\left(-\log\left(N/(1-\gamma N)\right)\right) = \frac{1-\gamma N}{N}.$$

Thus,

$$\begin{aligned}
\mathbb{P}\left(m_t \in N_t\middle|\mathcal{H}_t\right) &\geq 1 - \mathbb{P}\left(Y_j \leq -v^{-1}, \exists j \neq N_t\middle|\mathcal{H}_t\right) \\
&\geq 1 - \sum_{j \neq N_t} \mathbb{P}\left(Y_j < -v^{-1}\middle|\mathcal{H}_t\right) \\
&\geq 1 - (1-\gamma N) \\
&= \gamma N \\
&\geq 1 - \gamma.
\end{aligned}$$

The last inequality holds due to $\gamma \geq 1/(N+1)$. $\qquad\square$

## E   Proof of Theorem 3

*Proof.* Fix $t = 1, \ldots, T$ and let $V_t := \sum_{\tau=1}^t \sum_{i=1}^N X_i(\tau)X_i(\tau)^T + \lambda_t I$. For each $i$ and $\tau$, let $\widehat{\eta}_i(\tau) = Y_i^{DR}(\tau) - X_i(\tau)^T\beta$. Then

$$\widehat{\beta}_t = \beta + V_t^{-1}\left(-\lambda_t\beta + \sum_{\tau=1}^t \sum_{i=1}^N \widehat{\eta}_i(\tau)X_i(\tau)\right).$$

To bound $\left\|\widehat{\beta}_t - \beta\right\|_2$,

$$\begin{aligned}
\left\|\widehat{\beta}_t - \beta\right\|_2 &= \left\|V_t^{-1}\left(-\lambda_t\beta + \sum_{\tau=1}^t \sum_{i=1}^N \widehat{\eta}_i(\tau)X_i(\tau)\right)\right\|_2 \\
&\leq \left\|V_t^{-1}\right\|_2 \left\|\left(-\lambda_t\beta + \sum_{\tau=1}^t \sum_{i=1}^N \widehat{\eta}_i(\tau)X_i(\tau)\right)\right\|_2 \\
&= \{\lambda_{\min}(V_t)\}^{-1} \left\|\left(-\lambda_t\beta + \sum_{\tau=1}^t \sum_{i=1}^N \widehat{\eta}_i(\tau)X_i(\tau)\right)\right\|_2.
\end{aligned}$$

By Assumption 1, $\|\beta\|_2 \leq 1$. Using triangle inequality,

$$\left\|\widehat{\beta}_t - \beta\right\|_2 \leq \{\lambda_{\min}(V_t)\}^{-1}\lambda_t + \{\lambda_{\min}(V_t)\}^{-1}\left\|\sum_{\tau=1}^t \sum_{i=1}^N \widehat{\eta}_i(\tau)X_i(\tau)\right\|_2. \qquad (19)$$

We will bound the first term in (19). Let $\text{Tr}(A)$ be the trace of a matrix $A$. By the definition of the Frobenious norm, for $\tau = 1, \ldots, t$, and for $i = 1, \ldots, N$,

$$\left\|\sum_{i=1}^N X_i(\tau)X_i(\tau)^T\right\|_F \leq \sum_{i=1}^N \sqrt{\text{Tr}\left(X_i(\tau)X_i(\tau)^T X_i(\tau)X_i(\tau)^T\right)} \leq N.$$

By Assumptions 3 and 4, $\left\{\sum_{i=1}^{N} X_i(\tau) X_i(\tau)^T\right\}_{\tau=1}^{t}$ are independent random variables such that $\mathbb{E}\left[\sum_{i=1}^{N} X_i(\tau) X_i(\tau)^T\right] \geq N\phi^2 > 0$. Let $\delta \in (0,1)$ be given. By Lemma 6, if we set $\lambda_t = 4\sqrt{2}N\sqrt{t \log \frac{12t^2}{\delta}}$,

$$\{\lambda_{\min}(V_t)\}^{-1} < \frac{1}{\phi^2 N t},$$

holds with probability at least $1 - \delta/(3t^2)$. Thus, the first term can be bounded by

$$\{\lambda_{\min}(V_t)\}^{-1}\lambda_t \leq \frac{4\sqrt{\log \frac{12t^2}{\delta}}}{\sqrt{t}\phi^2}. \tag{20}$$

Now we will bound the second term in (19). Let $U_i(\tau) := X_i(\tau) X_i(\tau)^T (\breve{\beta}_\tau - \beta)$. Then we can decompose $\widehat{\eta}_i(\tau) X_i(\tau)$ as,

$$
\begin{aligned}
\widehat{\eta}_i(\tau) X_i(\tau) &= U_i(\tau) + \frac{\mathbb{I}(a_\tau = i)}{\pi_i(\tau)}\left(Y_i(\tau) - X_i(\tau)^T \breve{\beta}_\tau\right) X_i(\tau) \\
&= \left(1 - \frac{\mathbb{I}(a_\tau = i)}{\pi_i(\tau)}\right) U_i(\tau) + \frac{\mathbb{I}(a_\tau = i)}{\pi_i(\tau)} \eta_i(\tau) X_i(\tau) \\
&:= D_i(\tau) + E_i(\tau).
\end{aligned}
\tag{21}
$$

Let $D_\tau := \sum_{i=1}^{N} D_i(\tau)$. Since $U_i(\tau)$ is $\mathcal{H}_\tau$-measurable, the conditional expectation of $D_\tau$ is

$$
\begin{aligned}
\mathbb{E}\left[D_\tau \mid \mathcal{H}_\tau\right] &= \mathbb{E}\left[\sum_{i=1}^{N} D_i(\tau) \mid \mathcal{H}_\tau\right] = \sum_{i=1}^{N} \mathbb{E}\left[\left(1 - \frac{\mathbb{I}(a_\tau = i)}{\pi_i(\tau)}\right) \mid \mathcal{H}_\tau\right] U_i(\tau) \\
&= \sum_{i=1}^{N}\left(1 - \frac{\pi_i(\tau)}{\pi_i(\tau)}\right) U_i(\tau) = 0
\end{aligned}
$$

Thus, $\left\{\sum_{u=1}^{\tau} D_u\right\}_{\tau=1}^{t}$ is a martingale sequence on $\left(\mathbb{R}^d, \|\cdot\|_2\right)$ with respect to $\mathcal{H}_\tau$. By Lemma 10, since $\left(\mathbb{R}^d, \|\cdot\|_2\right)$ is a Hilbert space, there exists a martingale sequence $\{P_\tau\}_{\tau=1}^{t} = \left\{\left(P_\tau^{(1)}, P_\tau^{(2)}\right)^T\right\}_{\tau=1}^{t}$ on $\mathbb{R}^2$ such that

$$\left\|\sum_{u=1}^{\tau} D_u\right\|_2 = \|P_\tau\|_2, \quad \|D_\tau\|_2 = \|P_\tau - P_{\tau-1}\|_2 \tag{22}$$

and $P_0 = 0$, for any $\tau = 1, \ldots, t$. Since $\left\|\breve{\beta}_\tau - \beta\right\|_2 \leq b$, for $r = 1, 2$

$$
\begin{aligned}
\left|P_\tau^{(r)} - P_{\tau-1}^{(r)}\right| &\leq \|P_\tau - P_{\tau-1}\|_2 = \|D_\tau\|_2 \\
&= \left\|\sum_{i=1}^{N}\left(1 - \frac{\mathbb{I}(a_\tau = i)}{\pi_i(\tau)}\right) U_i(\tau)\right\| \\
&\leq \sum_{i=1}^{N}\left|1 - \frac{\mathbb{I}(a_\tau = i)}{\pi_i(\tau)}\right| \|U_i(\tau)\|_2 \\
&\leq \sum_{i=1}^{N}\left|1 - \frac{\mathbb{I}(a_\tau = i)}{\pi_i(\tau)}\right| \left\|\breve{\beta}_\tau - \beta\right\|_2 \\
&\leq \left(N - 1 + \frac{1}{\pi_{a_\tau}(\tau)} - 1\right) b \\
&\leq \left(N + \pi_{a_\tau}(\tau)^{-1}\right) b.
\end{aligned}
$$

By Lemma 11, there exists a martingale sequence $\left\{N_\tau^{(r)}\right\}_{\tau=1}^t$ such that $\left|N_\tau^{(r)} - N_{\tau-1}^{(r)}\right| \leq (N + \gamma^{-1})b$, for all $\tau = 1, \ldots, t$ and

$$\left\{N_t^{(r)} \neq P_t^{(r)}\right\} \subset \bigcup_{\tau=1}^t \left\{\left|P_\tau^{(r)} - P_{\tau-1}^{(r)}\right| > (N + \gamma^{-1})b\right\} \subset \bigcup_{\tau=1}^t \left\{\pi_{a_\tau}(\tau) \leq \gamma\right\}. \tag{23}$$

Thus, by (22) and (23), for any $x > 0$,

$$\begin{aligned}
\mathbb{P}\left(\left\|\sum_{u=1}^t D_u\right\|_2 > x, \bigcap_{\tau=1}^T \{\pi_{a_\tau}(\tau) > \gamma\}\right) &= \mathbb{P}\left(\|P_t\|_2 \geq x, \bigcap_{\tau=1}^T \{\pi_{a_\tau}(\tau) > \gamma\}\right) \\
&\leq \mathbb{P}\left(\sum_{r=1}^2 \left|P_t^{(r)}\right| \geq x, \bigcap_{\tau=1}^t \{\pi_{a_\tau}(\tau) > \gamma\}\right) \\
&\leq \sum_{r=1}^2 \mathbb{P}\left(\left|P_t^{(r)}\right| \geq \frac{x}{2}, \bigcap_{\tau=1}^t \{\pi_{a_\tau}(\tau) > \gamma\}\right) \\
&\leq \sum_{r=1}^2 \mathbb{P}\left(\left|P_t^{(r)}\right| \geq \frac{x}{2}, N_t^{(r)} = P_t^{(r)}\right) \\
&\leq \sum_{r=1}^2 \mathbb{P}\left(\left|N_t^{(r)}\right| \geq \frac{x}{2}\right).
\end{aligned}$$

Since $N_\tau^{(r)}$ has bounded differences, we can apply Lemma 9 to have

$$\sum_{r=1}^2 \mathbb{P}\left(\left|N_t^{(r)}\right| \geq \frac{x}{2}\right) \leq 4\exp\left(-\frac{x^2}{8tb^2(N+\gamma^{-1})^2}\right).$$

Thus, with probability at least $1 - \delta/(3t^2)$,

$$\left\|\sum_{\tau=1}^t D_\tau\right\|_2 \leq 2\sqrt{2}b(N + \gamma^{-1})\sqrt{\log\frac{12t^2}{\delta}} \tag{24}$$

holds with the event $\bigcap_{t=1}^T \{\pi_{a_t}(t) > \gamma\}$.

Now we will bound the $E_i(\tau)$ term in (21). Under the event $\bigcap_{t=1}^T \{\pi_{a_t}(t) > \gamma\}$, we have

$$\sum_{\tau=1}^t \sum_{i=1}^N E_i(\tau) = \sum_{\tau=1}^t \frac{\eta_{a_\tau}(\tau)}{\pi_{a_\tau}(\tau)} X_{a_\tau}(\tau) = \sum_{\tau=1}^t \frac{\mathbb{I}(\pi_{a_t}(t) > \gamma)\,\eta_{a_\tau}(\tau)}{\pi_{a_\tau}(\tau)} X_{a_\tau}(\tau)$$

For each $\tau \geq 1$, define a filtration $\mathcal{F}_{\tau-1} := \mathcal{H}_\tau \cup \{a_\tau\}$. Then $X_{a_\tau}(\tau)$ is $\mathcal{F}_{\tau-1}$-measurable. By Assumption 2, for any $\lambda \in \mathbb{R}$,

$$\mathbb{E}\left[\exp\left(\lambda \frac{\mathbb{I}(\pi_{a_t}(t) > \gamma)\,\eta_{a_\tau}(\tau)}{\pi_{a_\tau}(\tau)}\right)\bigg| \mathcal{F}_{\tau-1}\right] \leq \exp\left(\frac{\lambda^2 \mathbb{I}(\pi_{a_t}(t) > \gamma)\,\sigma^2}{2\pi_{a_\tau}(\tau)^2}\right) \leq \exp\left(\frac{\lambda^2\sigma^2}{2\gamma^2}\right),$$

almost surely. Since $\|X_{a_\tau}(\tau)\|_2 \leq 1$, by Lemma 4, there exists an absolute constant $C > 0$ such that, with probability at least $1 - \delta/(3t^2)$,

$$\left\|\sum_{\tau=1}^t \sum_{i=1}^N E_i(\tau)\right\|_2 \leq 2C\sigma\gamma^{-1}\sqrt{t}\sqrt{\log\frac{12t^2}{\delta}}. \tag{25}$$

Thus, with (20), (24), and (25), under the event $\bigcap_{t=1}^{T} \{\pi_{a_t}(t) > \gamma\}$, we have

$$
\begin{aligned}
\left\|\widehat{\beta}_t - \beta\right\|_2 &\leq \frac{4\sqrt{\log \frac{12t^2}{\delta}}}{\sqrt{t}\phi^2} + \frac{1}{\phi^2 Nt}\left(4\left(N + \gamma^{-1}\right)b\sqrt{t}\sqrt{\log \frac{12t^2}{\delta}} + 2C\sigma\gamma^{-1}\sqrt{t}\sqrt{\log \frac{12t^2}{\delta}}\right) \\
&\leq \frac{4 + 4b + \gamma^{-1}N^{-1}\left(4b + 2C\sigma\right)}{\phi^2\sqrt{t}}\sqrt{\log \frac{12t^2}{\delta}} \\
&\leq \frac{4 + 4b + 2\left(4b + 2C\sigma\right)}{\phi^2\sqrt{t}}\sqrt{\log \frac{12t^2}{\delta}} \\
&:= \frac{C_{b,\sigma}}{\phi^2\sqrt{t}}\sqrt{\log \frac{12t^2}{\delta}},
\end{aligned}
\tag{26}
$$

with probability at least $1 - \delta/t^2$. Since (26) holds for all $t = 1, \ldots, T$,

$$
\begin{aligned}
&\leq \mathbb{P}\left(\bigcup_{t=1}^{T}\left\{\left\|\widehat{\beta}_t - \beta\right\|_2 > \frac{C_{b,\sigma}}{\phi^2\sqrt{t}}\sqrt{\log \frac{12t^2}{\delta}}\right\}, \bigcap_{t=1}^{T}\{\pi_{a_t}(t) > \gamma\}\right) \\
&\leq \mathbb{P}\left(\bigcup_{t=1}^{T}\left\{\left\|\widehat{\beta}_t - \beta\right\|_2 > \frac{C_{b,\sigma}}{\phi^2\sqrt{t}}\sqrt{\log \frac{12t^2}{\delta}}\right\}, \bigcap_{t=1}^{T}\{\pi_{a_t}(t) > \gamma\}\right) \\
&\leq \sum_{t=1}^{T}\mathbb{P}\left(\left\|\widehat{\beta}_t - \beta\right\|_2 > \frac{C_{b,\sigma}}{\phi^2\sqrt{t}}\sqrt{\log \frac{12t^2}{\delta}}, \bigcap_{t=1}^{T}\{\pi_{a_t}(t) > \gamma\}\right) \\
&\leq \delta.
\end{aligned}
$$

$\square$

# F    Proof of Lemma 4

*Proof.* Fix a $t \geq 1$. Since for each $\tau = 1, \ldots, t$, $\mathbb{E}\left[\eta(\tau)| \mathcal{F}_{\tau-1}\right] = 0$ and $X(\tau)$ is $\mathcal{F}_{\tau-1}$-measurable, the stochastic process,

$$
\left\{\sum_{\tau=1}^{u}\eta(\tau)X(\tau)\right\}_{u=1}^{t}
\tag{27}
$$

is a $\mathbb{R}^d$-martingale. Since $(\mathbb{R}^d, \|\cdot\|_2)$ is a Hilbert space, by Lemma 10, there exists a $\mathbb{R}^2$-martingale $\{M_u\}_{u=1}^{t}$ such that

$$
\left\|\sum_{\tau=1}^{u}\eta(\tau)X(\tau)\right\|_2 = \|M_u\|_2, \quad \|\eta(u)X(u)\|_2 = \|M_u - M_{u-1}\|_2,
\tag{28}
$$

and $M_0 = 0$. Set $M_u = (M_1(u), M_2(u))^T$. Then for each $i = 1, 2$, and $u \geq 2$, by the assumption $\|X(u)\|_2 \leq 1$,

$$
\begin{aligned}
|M_i(u) - M_i(u-1)| &\leq \|M_u - M_{u-1}\|_2 \\
&= \|\eta(u)X(u)\|_2 \\
&\leq |\eta(u)|.
\end{aligned}
$$

By Lemma 12, $M_i(u) - M_i(u-1)$ is $C\sigma$-sub-Gaussian for some constant $C > 0$. By Lemma 9, for $x > 0$,

$$
\begin{aligned}
\mathbb{P}\left(|M_i(t)| > x\right) &= \mathbb{P}\left(\left|\sum_{u=1}^{t}M_i(u) - M_i(u-1)\right| > x\right) \\
&\leq 2\exp\left(-\frac{x^2}{2tC^2\sigma^2}\right),
\end{aligned}
$$

for each $i = 1, 2$. Thus, with probability $1 - \delta/(2t^2)$,

$$M_i(t)^2 \leq 2tC^2\sigma^2 \log \frac{4t^2}{\delta}.$$

In summary, with probability at least $1 - \delta/t^2$,

$$\left\| \sum_{\tau=1}^{t} \eta(\tau)X(\tau) \right\|_2 = \sqrt{M_1(t)^2 + M_2(t)^2} \leq 2C\sigma\sqrt{t}\sqrt{\log \frac{4t^2}{\delta}}.$$

$\square$

# G   Proof of Lemma 6

*Proof.* For each $\tau = 1, \ldots, t$, let $\Sigma_\tau = \mathbb{E}\left[P(\tau)|\, \mathcal{F}_{\tau-1}\right]$. Since $P(\tau)$ and $\Sigma_\tau$ are symmetric matrices,

$$
\begin{aligned}
\lambda_{\min}\left( \sum_{\tau=1}^{t} P(\tau) + \lambda_t I \right) =& \lambda_{\min}\left( \sum_{\tau=1}^{t} P(\tau) \right) + \lambda_t \\
=& \lambda_{\min}\left( \sum_{\tau=1}^{t} \{P(\tau) - \Sigma_\tau\} + \sum_{\tau=1}^{t} \Sigma_\tau \right) + \lambda_t \\
\geq& \lambda_{\min}\left( \sum_{\tau=1}^{t} \{P(\tau) - \Sigma_\tau\} \right) + \sum_{\tau=1}^{t} \lambda_{\min}(\Sigma_\tau) + \lambda_t \\
\geq& \lambda_{\min}\left( \sum_{\tau=1}^{t} \{P(\tau) - \Sigma_\tau\} \right) + \phi^2 t + \lambda_t.
\end{aligned}
$$

The last inequality uses the fact that $\lambda_{\min}(\Sigma_\tau) \geq \phi^2$ for all $\tau$.

$$
\begin{aligned}
\mathbb{P}\left( \lambda_{\min}\left( \sum_{\tau=1}^{t} P(\tau) + \lambda_t I \right) \leq \phi^2 t \right) \leq& \mathbb{P}\left( \lambda_{\min}\left( \sum_{\tau=1}^{t} \{P(\tau) - \Sigma_\tau\} \right) + \lambda_t \leq 0 \right) \\
=& \mathbb{P}\left( \lambda_{\max}\left( \sum_{\tau=1}^{t} \{\Sigma_\tau - P(\tau)\} \right) \geq \lambda_t \right) \qquad (29) \\
\leq& \mathbb{P}\left( \left\| \sum_{\tau=1}^{t} \{\Sigma_\tau - P(\tau)\} \right\|_F \geq \lambda_t \right).
\end{aligned}
$$

Set $S_u = \sum_{\tau=1}^{u} \{\Sigma_\tau - P(\tau)\}$. Then $\{S_u\}_{u=1}^{t}$ can be regarded as a martingale sequence on $\mathbb{R}^{d \times d}$ with respect to $\{P(\tau)\}_{\tau=1}^{t}$. Note that $\left( \mathbb{R}^{d \times d}, \|\cdot\|_F \right)$ is a Hilbert space. By Lemma 10, there exists a martingale sequence $\left\{ D_u = (D_1(u), D_2(u))^T \right\}_{u=1}^{t}$ on $\mathbb{R}^2$ such that

$$\|S_u\|_F = \sqrt{D_1(u)^2 + D_2(u)^2}, \quad \|M_u - \Sigma_u\|_F = \|D_u - D_{u-1}\|_2, \qquad (30)$$

for any $u \geq 1$, and $D_0 = 0$. Then, for any $i = 1, 2$,

$$|D_i(u) - D_i(u-1)|^2 \leq \|D_u - D_{u-1}\|_2^2 = \|P(u) - \Sigma_u\|_F^2$$

Since $\|P(u) - \Sigma_u\|_F \leq 2c$, we can apply Lemma 9 for $D_1(\tau)$, and $D_2(\tau)$, respectively. For any $i = 1, 2$, and for any $x > 0$,

$$\mathbb{P}\left( |D_i(t)| \geq x \right) \leq 2\exp\left( -\frac{x^2}{8c^2 t} \right).$$

From (29) and (30),

$$\mathbb{P}\left(\lambda_{\min}\left(\sum_{\tau=1}^{t} P(\tau) + \lambda_t I\right) \leq \phi^2 t\right) \leq \mathbb{P}\left(\|S_t\|_F \geq \lambda_t\right)$$

$$= \mathbb{P}\left(\sqrt{D_1(t)^2 + D_2(t)^2} \geq \lambda_t\right)$$

$$\leq \mathbb{P}\left(|D_1(t)| + |D_2(t)| \geq \lambda_t\right)$$

$$\leq \mathbb{P}\left(|D_1(t)| \geq \frac{\lambda_t}{2}\right) + \mathbb{P}\left(|D_2(t)| \geq \frac{\lambda_t}{2}\right)$$

$$\leq 4\exp\left(-\frac{\lambda_t^2}{32c^2 t}\right).$$

Thus, for any $\delta \in (0,1)$, if $\lambda_t \geq 4\sqrt{2}c\sqrt{t}\sqrt{\log\frac{4t^2}{\delta}}$, then with probability at least $1-\delta$,

$$\lambda_{\min}\left(\sum_{\tau=1}^{t} P(\tau) + \lambda_t I\right) > \phi^2 t.$$

$\square$

## H   Implementation details

### H.1   Efficient calculation of the sampling probability

In our proposed algorithm, we use quasi-Monte Carlo estimation to calculate the sampling probability, $\tilde{\pi}_i(t)$. At round $t$, for each $i = 1, \ldots, N$, define $Z_i = \frac{X_i(t)^T\left(\tilde{\beta}_i(t) - \widehat{\beta}_{t-1}\right)}{v\|X_i(t)\|_{V_t^{-1}}}$. Then, $Z_1, \ldots, Z_N$ are IID standard Gaussian random variables. For each $i = 1, \ldots, N$,

$$\tilde{\pi}_i(t) = \mathbb{P}\left(X_i(t)^T\tilde{\beta}_i(t) \geq X_j(t)^T\tilde{\beta}_j(t), \forall j \neq i \,\middle|\, \mathcal{H}_t\right)$$

$$= \mathbb{P}\left(\frac{\|X_i(t)\|_{V_t^{-1}}}{\|X_j(t)\|_{V_t^{-1}}}Z_i \geq Z_j + \frac{(X_j(t) - X_i(t))^T\widehat{\beta}_{t-1}}{v\|X_j(t)\|_{V_t^{-1}}}, \forall j \neq i \,\middle|\, \mathcal{H}_t\right)$$

let $f$ and $F$ be the density and the distribution function of the standard Gaussian random variables, respectively. Since $Z_i$, and $\{Z_j\}_{j \neq i}$ are independent, the selection probability can be written as,

$$\tilde{\pi}_i(t) = \int \prod_{j \neq i} F\left(\frac{\|X_i(t)\|_{V_t^{-1}}}{\|X_j(t)\|_{V_t^{-1}}}z + \frac{(X_i(t) - X_j(t))^T\widehat{\beta}_{t-1}}{v\|X_j(t)\|_{V_t^{-1}}}\right)f(z)dz.$$

This can be estimated by,

$$\frac{1}{M}\sum_{m=1}^{M} F\prod_{j \neq i}\left(\frac{\|X_i(t)\|_{V_t^{-1}}}{\|X_j(t)\|_{V_t^{-1}}}Z^{(m)} + \frac{(X_i(t) - X_j(t))^T\widehat{\beta}_{t-1}}{v\|X_j(t)\|_{V_t^{-1}}}\right), \tag{31}$$

where $Z^{(m)}$ is the standard Gaussian random variables.

In this way, we can compute $\tilde{\pi}_i(t)$ without sampling $\tilde{\beta}_i(t)$ $M \times N$ times from $N(\widehat{\beta}_{t-1}, v_t I)$. The error of the quasi Monte Carlo method is bounded by $O\left(\frac{(\log M)^s}{M}\right)$, where $s$ is the dimension of the domain of function to integrate. If we sample $\tilde{\beta}_i(t)$ $M \times N$ times, it gives $O\left(\frac{(\log M)^{N-1}}{M}\right)$ error. In contrast, using (31) reduces the error to $O\left(\frac{\log M}{M}\right)$.

In our simulation studies, we use `sobol_seq` module in Python 3 to generate the quasi-Monte Carlo samples. The number of samples is $M = 200$ in BLTS and DRTS. We plot the estimator of $\tilde{\pi}_i(t)$ using $m = 1, \ldots, 200$ quasi-Monte Carlo samples, and observe that it converges within the small errors.

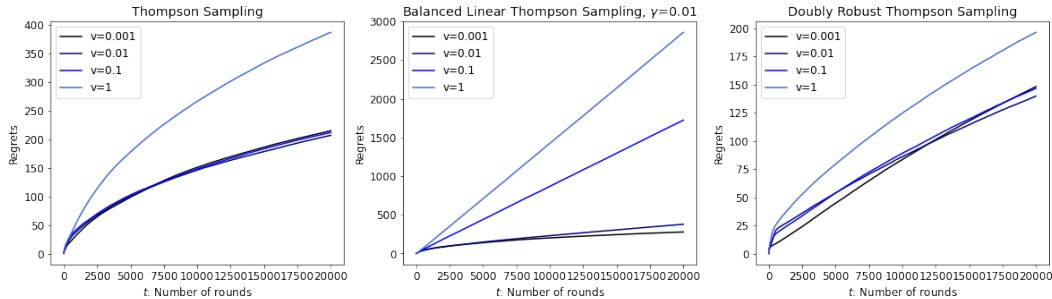

Figure 2: A comparison of the cumulative regrets of `LinTS` (left), `BLTS` (middle), and `DRTS` (right) on various $v$ when $d = 20$ and $N = 20$. Each line shows the averaged cumulative regrets over 10 repeated experiments.

## H.2 Simulation results with various hyperparameters

In this subsection, we report the performance of the three algorithms, (i) `LinTS`, (ii) `BLTS`, and (iii) the proposed `DRTS`, with various hyperparameters. As described in Section 5, the hyperparameter sets are $v \in \{0.001, 0.01, 0.1, 1\}$ for all three algorithms and $\gamma \in \{0.01, 0.5, 0.1\}$ for `BLTS`.

Figure 2 shows the comparison of the three algorithms on various hyperparameter $v$, when $d = 20, N = 20, \gamma = 0.01$. We find that the performance of the three algorithms do not change much when $v \leq 0.01$. This trend is similar on different $\gamma, N$, and $d$.

# I   A review of approaches to missing data and doubly-robust method

In this section, we review approaches to missing data and the doubly-robust method used in our proposed method. First, we provide the approaches from a purely missing data point of view and how the doubly-robust method is motivated. In the second section, we show the procedures applying the doubly-robust method in bandit settings.

## I.1   Doubly-robust method in missing data

There are two main approaches to missing data: imputation and inverse probability weighting (IPW). Imputation is to fill in the predicted value of missing data from a specified model, and IPW is to use the observed records only but weight them by the inverse of the observation probability. The doubly-robust method can be viewed as a combination of the two.

For illustrative purposes, consider the problem of estimating the marginal mean of $Y \in \mathbb{R}$, $\mathbb{E}(Y) =: \mu$. Denoting $(Y_i - \mu)$ by $U_i(\mu)$, when all data are observed,

$$U(\mu) = \sum_{i=1}^{n} U_i(\mu) = \sum_{i=1}^{n} (Y_i - \mu) = 0,$$

gives an unbiased estimator of $\mu$, $\sum_{i=1}^{n} Y_i/n$, and $U(\mu)$ is called an unbiased estimating function since $\mathbb{E}[U(\mu)] = 0$. Let $\delta_i$ be the observation indicator which takes value 1 if $Y_i$ is observed, 0, otherwise. Suppose there are auxiliary variables, $X_i \in \mathbb{R}^d$, and $X_i$'s are observed for all $i$. Also denote the probability of observation by $P(\delta_i = 1|X_i) =: \pi_i$. We assume $P(\delta_i = 1|Y_i, X_i) = P(\delta_i = 1|X_i)$, that is, the observation indicator is independent of $Y_i$. This is called *missing at random* mechanism. This assumption is required for the doubly robust method to be valid. Using the observed values only, the estimating equation for the observed data

$$U_o(\mu) = \sum_{i=1}^{n} \delta_i U_i(\mu) = \sum_{i=1}^{n} \delta_i(Y_i - \mu) = 0,$$

gives $\frac{\sum_{i=1}^{n} \delta_i Y_i}{\sum_{i=1}^{n} \delta_i}$ as an estimator for $\mu$. This estimator may be biased since $\mathbb{E}U_o(\mu) \neq 0$.

The two main approaches modify the observed estimating function employing two new quantities, $\mathbb{E}(Y_i|X_i)$ and $\pi_i$. These two quantities are usually unknown and we need to specify models. Therefore

the two approaches require assumptions for auxiliary models: the imputation model, $\mathbb{E}(Y_i|X_i;\beta)$, and the model for observation probability, $\pi_i(\phi)$. The validity of each approach depends on the correct specification of the auxiliary model assumptions. The qualifier 'auxiliary' comes from the fact that these models are not needed when there is no missing data. In IPW, one constructs an unbiased estimating equation by amplifying the observed record according to the inverse of the observation probability as follows:

$$\sum_{i=1}^{n} \frac{\delta_i}{\pi_i(\phi)} U_i(\mu) = \sum_{i=1}^{n} \frac{\delta_i}{\pi_i(\phi)} (Y_i - \mu).$$

If $\pi(\phi)$ is correctly specified, i.e., $\pi = \pi(\phi)$, $\mathbb{E}(\sum_{i=1}^{n} \frac{\delta_i}{\pi_i(\phi)} U_i(\mu)) = 0$, hence the resulting IPW estimator is valid. In the imputation method, we replace missing $Y_i$ with $\mathbb{E}(Y_i|X_i;\beta)$ and the estimator is the solution of $U^{IMP}(\mu,\beta) = 0$ where

$$U^{IMP}(\mu,\beta) = \sum_{i=1}^{n} [\delta_i U_i(\mu) + (1-\delta_i)\mathbb{E}(U_i(\mu)|X_i;\beta)]$$
$$= \sum_{i=1}^{n} [\mathbb{E}(Y_i|X_i;\beta) + \delta_i\{Y_i - \mathbb{E}(Y_i|X_i;\beta)\} - \mu].$$

The doubly robust (DR) method [Robins et al., 1994, Bang and Robins, 2005] was initially motivated by attempting to improve the efficiency of the IPW method. Note that we can construct an auxiliary unbiased estimating function $(\frac{\delta_i}{\pi_i(\phi)} - 1)$. Geometrically we can reduce the norm of the estimating function $\frac{\delta_i}{\pi_i(\phi)} U_i(\mu)$ by subtracting the projection on to the nuisance tangent space formed from $(\frac{\delta_i}{\pi_i(\phi)} - 1)$. The nuisance tangent space is the closed linear span of $B(\frac{\delta_i}{\pi_i(\phi)} - 1)$ for some $B \in \mathbb{R}^d$, and the projection onto the nuisance tangent space is

$$\sum_{i=1}^{n} \frac{\delta_i - \pi_i(\phi)}{\pi_i(\phi)} \mathbb{E}(U_i|X_i;\beta).$$

After subtraction, the DR estimating function has a form

$$U^{DR}(\mu,\beta,\phi) = \sum_{i=1}^{n} \left[ \frac{\delta_i}{\pi_i(\phi)} U_i(\mu) + (1 - \frac{\delta_i}{\pi_i(\phi)})\mathbb{E}(U_i|X_i;\beta) \right]$$
$$= \sum_{i=1}^{n} \left[ \mathbb{E}(U_i|X_i;\beta) + \frac{\delta_i}{\pi_i(\phi)} \{U_i(\mu) - \mathbb{E}(U_i(\mu)|X_i;\beta)\} \right].$$

Note that when you replace $\delta_i$ in $U^{IMP}(\mu)$ with $\frac{\delta_i}{\pi_i(\phi)}$, you obtain $U^{DR}(\mu)$. The DR method requires both auxiliary models. However, its validity is guaranteed when *either* of the models is correct. To verify, if the imputation model is correctly specified, i.e., $\mathbb{E}[U_i(\mu) - \mathbb{E}(U_i(\mu)|X_i;\beta)|X_i] = 0$, we have

$$\mathbb{E}\{U^{DR}(\mu,\beta,\phi)\} = \mathbb{E}\sum_{i=1}^{n} \left[ \mathbb{E}(U_i|X_i) - \frac{\delta_i}{\pi_i(\phi)} \{U_i(\mu) - \mathbb{E}(U_i(\mu)|X_i)\} \right] = \sum_{i=1}^{n} \mathbb{E}\mathbb{E}(U_i|X_i) = 0$$

even if the $\pi$ model is misspecified, i.e., $\pi_i(\phi) \neq \pi_i$. If the observation model is correctly specified, $\pi_i(\phi) = \pi_i$, then $\mathbb{E}(1 - \frac{\delta_i}{\pi_i}|X_i) = 0$, and

$$\mathbb{E}\{U^{DR}(\mu,\beta,\phi)\} = \sum_{i=1}^{n} \mathbb{E}\left[ \frac{\delta_i}{\pi_i} U_i(\mu) + \left\{ (1 - \frac{\delta_i}{\pi_i})\mathbb{E}(U_i|X_i;\beta) \right\} \right] = \sum_{i=1}^{n} \mathbb{E}\left[ \frac{\delta_i}{\pi_i} U_i(\mu) \right] = 0,$$

even if the imputation model is misspecified, i.e., $\mathbb{E}[U_i(\mu)|X_i] \neq \mathbb{E}[U_i(\mu)|X_i;\beta]$. Therefore when *either* of the models is correct, $U^{DR}(\mu)$ is unbiased and with other technical conditions, the estimator can be shown to be consistent. That is why the qualifier *doubly robust* is adopted. The construction of the DR estimating function is possible because we have two unbiased estimating functions.

## I.2  Application to bandit settings

In bandit settings, the missingness is controlled since the learner selects the arm. Therefore, the probability of observation or selection is known and the DR estimator is guaranteed to be valid although the imputation model for missing reward is incorrectly specified. The merit of the DR estimator in the bandit setting is that we can utilize the observed contexts from selected or unselected arms. Below we describe the DR method in the contextual bandit setting.

Let $\pi_i(t) := \mathbb{P}(a_t = i | \mathcal{H}_t)$ be the probability of selecting arm $i$ at round $t$. As defined in the manuscript, the DR pseudo-reward is

$$Y_i^{DR}(t) = \left\{ 1 - \frac{\mathbb{I}(i = a_t)}{\pi_i(t)} \right\} X_i(t)^T \breve{\beta}_t + \frac{\mathbb{I}(i = a_t)}{\pi_i(t)} Y_{a_t}(t), \tag{32}$$

for some $\breve{\beta}_t$ depending on $\mathcal{H}_t$. The pseudo-reward (32) comes from the following procedures. First we construct an unbiased estimating function also known as the IPW score,

$$\sum_{\tau=1}^{t} \sum_{i=1}^{N} \frac{\mathbb{I}(i = a_\tau)}{\pi_i(\tau)} X_i(\tau) \left( Y_i(\tau) - X_i(\tau)^T \beta \right), \tag{33}$$

where only the pairs $(X_i(t), Y_i(t))$ from the selected arms are contributed according the weight of the inverse of $\pi_i(t)$. Setting this score equal to 0 and solving $\beta$ gives the estimator used in Dimakopoulou et al. [2019]. Now we can subtract the projection on the nuisance tangent space from (33). The nuisance tangent space is the closed linear span of $B(\frac{\mathbb{I}(i=a(t))}{\pi_i(t)} - 1)$ for some $B \in \mathbb{R}^d$, and the projection onto the nuisance tangent space is

$$\sum_{\tau=1}^{t} \sum_{i=1}^{N} \frac{\mathbb{I}(i = a_\tau) - \pi_i(\tau)}{\pi_i(\tau)} X_i(\tau) \left( E(Y_i(\tau) | \mathcal{H}_\tau) - X_i(\tau)^T \beta \right).$$

When the projection is subtracted from the (33) after replacing $E(Y_i(t)|\mathcal{H}_t)$ with $X_i(t)^T \breve{\beta}_t$, the IPW score becomes the efficient score,

$$\sum_{\tau=1}^{t} \sum_{i=1}^{N} X_i(\tau) \left( Y_i^{DR}(\tau) - X_i(\tau)^T \beta \right). \tag{34}$$

Any $\breve{\beta}_t$ that depends on $\mathcal{H}_t$ serves the purpose of imputation. Due to the doubly robustness property, $X_i(t)^T \breve{\beta}_t$ does not have to be an unbiased estimator of $E(Y_i(t)|\mathcal{H}_t)$. We recommend setting $\breve{\beta}_t$ as the ridge regression estimator based on the selected arms only. The expression (34) resembles the score when the rewards for all arms were observed, if $Y_i(t)$ is replaced with $Y_i^{DR}(t)$.

Our proposed estimator $\widehat{\beta}_t$ is a solution of (34) with a regularization parameter $\lambda_t$:

$$\widehat{\beta}_t = \left( \sum_{\tau=1}^{t} \sum_{i=1}^{N} X_i(\tau) X_i(\tau)^T + \lambda_t I \right)^{-1} \left( \sum_{\tau=1}^{t} \sum_{i=1}^{N} X_i(\tau) Y_i^{DR}(\tau) \right).$$

Harnessing the pseudo-rewards defined in (32), we can make use of all contexts rather than just selected contexts. The use of all contexts instead of $X_{a_t}(t)$ induces the improvement in the regret bound of the proposed algorithm. Kim and Paik [2019] also suggests DR estimator, but it uses Lasso estimator from the following pseudo-reward

$$Y_i^{DR}(t) = \bar{X}(t)^T \hat{\beta}(t-1) + \frac{1}{N} \frac{Y_{a(t)}(t) - b_{a(t)}(t)^T \hat{\beta}(t-1)}{\pi_{a(t)}(t)},$$

where $\bar{X}(t) = \frac{1}{N} \sum_{i=1}^{N} X_i(t)$. This estimator is of an aggregated form. As described in the text, the estimator using the aggregated pseudo-reward does not permit the regret decomposition as equation (6) in the paper.

# J    Limitations of our work

1. The regret bound is constructed under the assumption that the contexts are independent over rounds (Assumption 3) and the covariance matrix is positive definite (Assumption 4). When using our proposed algorithm, one should check that the contexts satisfies the two assumptions. When the contexts violates the two assumptions, the improved regret bound (3) might not hold.

2. Our proposed algorithm, DRTS requires additional computations for $\tilde{\pi}_i(t)$, $\pi_i(t)$ and the imputation estimator $\check{\beta}$. To lessen this computational burdens we developed an efficient way to compute $\tilde{\pi}_i(t)$ and $\pi_i(t)$.