# OpenReview forum: "Doubly Robust Thompson Sampling with Linear Payoffs"
_NeurIPS.cc/2021/Conference — NeurIPS 2021 Spotlight_

### Official Review · Reviewer_SUDA · 2021-06-30

**Rating:** 7
**Confidence:** 3

**Summary:**

The paper proposes a new Thompson Sampling-based algorithm for the canonical N-armed T period contextual bandit problem that
achieves an expected cumulative regret of $\mathcal{O}\left( \phi^{-2}\sqrt{T} \right)$, where $\phi^2$ is the min eigenvalue of the context covariance matrix. It is easily observed that $\phi^{-2} \geqslant d$, where $d$ is the dimension of the context space. However, the authors point out that $\phi^{-2} = \mathcal{O}(d)$, in fact, holds in many practical scenarios. In such cases, therefore, the $\phi$-dependent scaling corresponds to an improvement upon the well-known $\tilde{\mathcal{O}}\left( d^{3/2}\sqrt{T}\right)$ standard by a factor of $\sqrt{d}$. The primary algorithmic innovation driving this result is the use of a "doubly robust" model for unobserved rewards, and a ridge estimator for the underlying $d$-dimensional (true) parameter vector, which facilitates a regret decomposition that is amenable to the aforementioned improvement in the upper bound. Numerical studies illustrate a significant performance improvement over standard well-known algorithms for the problem. The robust performance is plausibly due to a full utilization of the entire $N\times d \times T$ context stack, and use of the ridge estimator.

**Limitations And Societal Impact:**

The paper could benefit from a revision along the lines of the points raised above. I am willing to increase my score if these are (especially 4) adequately addressed.

**Main Review:**

The paper is well-written overall with accessible proofs. A provable $\sqrt{d}$-factor performance improvement in "typical" scenarios certainly makes it worthy of inclusion in the broader literature. Additionally, some of the technical development may be of independent interest in the analysis of related algorithms. Some broad remarks and typos are noted below:

1. Under a similar eigenvalue positivity condition per arm, can the algorithm and results be adapted to the setting where each arm has a different $\beta_i$? Or is it a non-trivial extension?

2. Table 1: Entry for Arm 3 at t=1 should be $Y_{a_1}(1)$.

3.  In Algorithm 1, inside the "for $l=1$ to $M_t$ do" loop, what if $\tilde{\pi}_{m_t}(t) \leqslant \gamma$ is always true? In this case, which arm $a_t$ is played after this loop has finished execution?

4. Line 181 - Theorem 1: How does the result depend on inputs to Algorithm 1, in particular, $M_t$? An appropriate specification of $M_t$ seems critical to the achievability of sub-linear regret. It is unclear how this is being ensured. Where is $M_t$ specified? Since the result is projected as the primary contribution, these points need to be appropriately addressed.



**Time Spent Reviewing:**

7

---

> ### Author Response · Authors · 2021-08-10
> **Responses to Reviewer SUDA**
>
> 1.**Q. Under a similar eigenvalue positivity condition per arm, can the algorithm and results be adapted to the setting where each arm has a different $\beta_i$? Or is it a non-trivial extension?**
>
> Thank you for this interesting question.
> It is a non-trivial extension since we need estimators for each $\beta_i$.  The main idea of our proposed method is to utilize the contexts from unselected arms.
> However, in this setting, these utilizable contexts do not contribute in estimating $\beta_i$.
>
> 2.**Table 1: Entry for Arm 3 at $t=1$ should be $Y_{a(1)}(1)$.**
>
> Thank you for the correction.
>
> 3.**In Algorithm 1, inside the "for $l=1$ to $M_t$ do" loop, what if $\tilde{\pi}_{m_t(t)}\le \gamma$ is always true? In this case, which arm at is played after this loop has finished execution?**
>
> Since $\gamma < 1/N$, there exists at least one arm $j$ such that $\tilde{\pi}\_j(t) > \gamma$.
> Thus with sufficiently large $M_t$, we can find $m_t$ such that $\tilde{\pi}\_{m_t}(t) > \gamma$.
> When the algorithm cannot find such $m_t$, the algorithm just chooses the $M_t$-th candidate arm.
> We will explicitly state this part in Algorithm 1 in the revision.
>
> 4.**Line 181 - Theorem 1: How does the result depend on inputs to Algorithm 1, in particular, $M_t$? An appropriate specification of $M_t$ seems critical to the achievability of sub-linear regret. It is unclear how this is being ensured. Where is $M_t$ specified? Since the result is projected as the primary contribution, these points need to be appropriately addressed.**
>
> Thank you for bringing up the issue of selecting $M_t$.
> Discussion on $M_t$ can be found in the Section A.3 in supplementary materials.
> Below we outline how we select $M_t$.
>
> The main point in selecting $M_t$ is to bound the probability that the resampling fails in finding an arm whose selection probability exceeds $\gamma$ for some $\delta$, i.e., $\mathbb{P}(a\_t\notin\\{i:\tilde{\pi}\_i(t) > \gamma\\})\le\delta/t^2$.
> We can bound the probability by choosing $M_t$ as a minimum integer that exceeds $\log\frac{t^2}{\delta}/\log\frac{1}{1-\gamma}$.
> Please see line 452 in supplementary materials.
> Intuitively, as $M_t$ increases, we have more opportunities for resampling and the probability that the resampling fails in finding arms from $\\{i:\tilde{\pi}_i(t)>\gamma\\}$ decreases.
> Since $\gamma < 1/N$, there exists $j$ such that $\tilde{\pi}\_j(t)>\gamma$, and the probability that the resampling fails is less than $1-\gamma$ in each resampling trial.
> Thus, by choosing $M_t$ for each round that satisfies $(1-\gamma)^{M_t}\le\delta/t^2$, the algorithm finds an arm $j$ such that $\tilde{\pi}_j(t)>\gamma$ in all rounds with high probability.
>
> Selecting an arm from the set $\\{i:\tilde{\pi}\_i(t) > \gamma\\}$ with high probability is crucial in achieving the regret bound of order $\tilde{O}(\phi^{-2}\sqrt{T})$  for two reasons.
> First, due to the relationship $\\{i:\tilde{\pi}\_i(t) > \gamma\\}\subseteq\\{i:\text{The arm } i\text{ is super-unsaturated}\\}$, it guarantees that the arm is super-unsaturated and our novel regret decomposition (6) holds to achieve our regret bound.
> Second, since $\pi_i(t) \ge \tilde{\pi}\_i(t)$, the inverse probability, $\pi_{a_t}(t)^{-1}$ is bounded by $\gamma^{-1}$ which appears in $Y^{DR}_i(t)$ and in the proof of Theorem 3.

---

> > ### Comment · Reviewer_SUDA · 2021-08-25
> > **Response to authors**
> >
> > I thank the authors for answering my queries. I am keeping my score of 7, and vote for an accept.

---

### Official Review · Reviewer_Qbta · 2021-07-11

**Rating:** 7
**Confidence:** 4

**Summary:**

This work proposes and analyzes a variant of the classical Thompson Sampling algorithm for linear contextual bandits (DRTS) which relies on a novel doubly robust estimator inspired from the missing data literature. They show a regret bound for their algorithm which depends on the context distribution and which, for a variety of different context distributions, improves on the state of the art for linear Thompson Sampling by a factor of \sqrt{d}. This result partially resolves a long-standing open question on the worst-case performance of linear Thompson Sampling.

**Limitations And Societal Impact:**

The authors do not discuss potential societal impact. However, the work is primarily theoretical so it is difficult to determine what negative societal impacts it may have. Limitations are discussed and addressed.

**Main Review:**

Thompson Sampling for the linear contextual bandit problem has been studied extensively. In the frequentist setting, the best known worst case upper bound for the classical linear Thompson Sampling algorithm (LinTS) scales as d^{3/2}\sqrt{T}, a factor of d^{1/2} off of the minimax optimal rate. The extra \sqrt{d} arises from inflating the covariance of the posterior by an extra factor of d to ensure sufficient exploration. It has been an open question whether this inflation factor of d can be removed for LinTS and the regret improved to d\sqrt{T} (see chapter 36 of [1]).

This work proposes a variant of the LinTS algorithm which relies on a novel estimator inspired by the the missing data literature. The authors show that their algorithm does not require inflating the covariance of the posterior by a factor of d. As such, instead of incurring regret of d^{3/2}\sqrt{T}, they show that they are able to obtain regret scaling as a function of the context distribution, which they show that, in many cases, yields a regret of d\sqrt{T}. While this does not completely resolve the open question around LinTS, it takes a major step forward in illustrating how the context distribution can be leveraged to obtain tighter regret bounds.


Pros:
- The improvement of the regret bound for Thompson Sampling to d\sqrt{T} in some cases is a noteworthy contribution. Furthermore, the explicit dependence on the context distribution in the regret bound is interesting and, to my knowledge, novel in existing minimax results. As noted above, the improvements here partially answer an open question and are a major step forward in our understanding of Thompson Sampling for linear bandits.
- The proposed estimator of \beta is novel in the bandits literature, to my knowledge, and could potentially be used to improve dimension dependence in other linear bandit algorithms.
- The dimension-free estimation error bound (Theorem 3) is a novel contribution itself and may be of independent interest.
- Simulation results are also convincing as to the effectiveness of DRTS.

Cons:
- While in some cases it is better, the regret bound could actually be worse than the existing d^{3/2} \sqrt{T} regret bounds for LinTS. Is it possible to instead get regret of \min{d^{3/2}, \phi^{-2}}? For really applications of contextual linear bandits, the context distribution could be nearly anything (e.g. the feature vectors corresponding to a stream of users we have no control over) so without some sort of a worst-case guarantee, this regret bound could actually be quite bad. Can the \phi be improved from being the worst-case lower bound on the minimum eigenvalue to, e.g., the average case lower bound on the minimum eigenvalue?
- Alternatively, if one could show a lower bound that depended on \phi when running a TS-like algorithm without inflating the covariance by a factor of d, this would also help in convincing the reader that the \phi is fundamental and necessary.
- More discussion and examples should be given on the value of \phi obtained by various context distributions. While there is a short paragraph after Theorem 1 giving some discussion of \phi, I would like to see more precise statements of the value of \phi in the various settings mentioned.
- \check{\beta} (the imputation estimator) is not specified in the main text. This should be defined. Furthermore, it should be clarified what the value of C_{b,\sigma} is and what value of b is achievable  if \| \check{\beta} - \beta \|_2 \le b. From the appendix, it seems C_{b,\sigma} = O(b), so it should be shown that b does not need to scale with d (naively one might imagine we would have b = \Omega(\sqrt{d})).
- As stated, the algorithm cannot be run in a computationally efficient way, due to the computation of \tilde{\pi}_t. While an approximation to this is proposed in the appendix, it is not stated how using the approximate version affects the theoretical results. Furthermore, it is not stated how to compute \pi_t (which is needed to compute Y^{DR}(t)).
- It would be interesting to see an experimental comparison against the version of LinTS which does not have covariance inflated by a factor of d. While this version doesn’t have theoretical guarantees in the frequentist setting, it is known to often perform well empirically, and it would be interesting to see how it compares to DRTS.

Overall, this work makes a non-trivial contribution to our understanding of Thompson Sampling in the linear contextual bandit setting, partially answering an open question, and, as such, I believe merits an accept, assuming the issues outlined above are resolved.

[1] Lattimore, Tor, and Csaba Szepesvári. Bandit algorithms. Cambridge University Press, 2020.

**Time Spent Reviewing:**

2

---

> ### Author Response · Authors · 2021-08-10
> **Responses to Reviewer Qbta**
>
> Cons 1.**Is it possible to instead get regret of $\min\\{d^{3/2}, \phi^{-2}\\}$?**
>
> Thank you for this question.
> To prove the desired regret bound, we need to obtain a bound for
> $||\widehat{\beta}_t-\beta||\_{V\_t}$, which is not a straightforward derivation and we are working on it.
>
> **Q. Can the $\phi$ be improved from being the worst-case lower bound on the minimum eigenvalue to, e.g., the average case lower bound on the minimum eigenvalue?**
>
> In our paper, we derive the minimum eigenvalue of the empirical gram matrix is bounded below for all $t$ based on the assumption on the minimum eigenvalue of the expectation of the gram matrix.
> We will think about how this can be done in an alternative way.
>
> Cons 2.**Obtaining a lower bound in terms of $\phi$:**
>
> This also is an excellent point.
> We will put the related analysis in the future work.
>
> Cons 3. **More discussion and examples should be given on the value of $\phi$ obtained by various context distributions:**.
>
> We will add the following discussion and examples on the revised version.
> When the context has uniform distribution on the unit ball, $\phi^{-2}=d+2$.
> When the context has truncated multivariate normal distribution with mean 0 and covariance $\Sigma$, we have $\phi^{-2} = (d+2)\exp(\frac{1}{2\lambda_{\min}(\Sigma)})$.
> For more examples, we refer to [1].
> More generally, if the density of contexts $p_X(x)$ satisfies $0< p_{\min} <p_X(x) < p_{\max}$ for all $x$ in the unit ball, then we have $\phi^{-2}=(d+2)\frac{p_{\max}}{p_{\min}}$.
>
> Cons 4.**$\check{\beta}$ (the imputation estimator) is not specified in the main text. This should be defined. Furthermore, it should be clarified what the value of $C\_{b,\sigma}$ is and what value of $b$ is achievable if $|| \check{\beta} - \beta ||\_2 \le b$. From the appendix, it seems $C_{b,\sigma} = O(b)$, so it should be shown that $b$ does not need to scale with $d$ (naively one might imagine we would have $b = \Omega(\sqrt{d}))$.**
>
> In summary, the condition, $|| \check{\beta}\_t - \beta ||\_2 \le b$, where $b=O(1)$, is achievable.
> We divide the discussion in two cases where (i) $\check{\beta}\_t$ is a consistent estimator such as ridge estimator used in our experiment, and (ii) $\check{\beta}\_t$ is a biased imputation estimator.
> As the reviewer pointed out, our regret depends on $C_{b,\sigma}$ where  $C_{b,\sigma} = \frac{4+4b + 2(4b+2C\sigma)}{\phi^2}$.   Note that $b$ is linear in $C_{b,\sigma}$.
> When $\check{\beta}\_t$ is a consistent estimator such as a ridge estimator which we described in line 309-310 in Section 5, we have $b=\tilde{O}(d/\sqrt{t})$.
> Then $C_{b,\sigma}$ can be partitioned into $C_{b,\sigma}= C_{\sigma}^* + \phi^{-2}\tilde{O}(d/\sqrt{t})$ where $C_{\sigma}^* = \frac{4 + 2(2C\sigma)}{\phi^2}$ is the main order term.
> Therefore the dependence of $\check{\beta}\_t$ on $d$ does not affect the order of $C\_{b,\sigma}$.
>
> When $\check{\beta}_t$ is not consistent, we can rescale or project the estimator so that the norm is bounded by a constant $C$, i.e., $||\check{\beta}_t||\_2 \le C$.
> In this way, we have $||\check{\beta}\_t-\beta||\_2 \le ||\check{\beta}\_t||\_2+||\beta||\_2 \le C+1$.
> Please also see **response for all reviewers** for using a biased imputation estimator and a doubly-robust inference.
>
> Cons 5. **While an approximation to this is proposed in the appendix, it is not stated how using the approximate version affects the theoretical results. Furthermore, it is not stated how to compute $\pi_t$ (which is needed to compute $Y^{DR}(t)$).**
>
> The formula for $\pi_{i}(t)$ is presented in Section A.2, (15) and (16).
> By plugging in the Monte-Carlo estimate $\hat{\tilde{\pi}}_i(t)$ of ${\tilde{\pi}_i(t)}$ in computing $\pi_i(t)$ via (15) and (16), we obtain a consistent estimate of $\pi_i(t).$
> The estimate of $\tilde{\pi}_i(t)$ is required in (i) computing the pseudo-reward $Y_i^{DR}(t)$ and (ii) deciding whether to resample or not.
> We show below that for both cases, replacing $\tilde{\pi}_i(t)$ with $\hat{\tilde{\pi}}_i(t)$ does not affect the order of the regret bound.
> Regarding case (i), let $\hat{Y}_i^{DR}(t)$ be the pseudo-reward with $\hat{\tilde{\pi}}_i(t)$ plugged-in.
> Replacing $\hat{Y}_i^{DR}(t)$ with $Y_i^{DR}(t)$ produces extra terms that are multiples of $\{\hat{\tilde{\pi}}_i(t)-\tilde{\pi}_i(t)\}$, which is of order  $\sqrt{1/M}$, where $M$ is the number of Monte-Carlo samples.
> We can make this term arbitrarily small by choosing sufficiently large $M$.
> Consequently, the regret bound has the same order as the current regret bound.
> As for case (ii), we note that the difference is negligible.
> The event $\\{\hat{\tilde{\pi}}\_i(t) > \gamma\\}$ means that $\\{{\tilde{\pi}}\_i(t) > \gamma-O(\sqrt{1/M})\\}$ with high probability. Let $\gamma'=\gamma-O(\sqrt{1/M}).$ If $\gamma\in (1/(N+1), 1/N)$ and $M$ is sufficiently large, we have $\gamma' \in  (1/(N+1), 1/N)$ as well and consequently, the regret bound has the same order as the current bound.
> These results are coherent with those in missing data literature in that replacing $\pi$ with its consistent estimator preserves asymptotic unbiasedness of the estimator.
>
> Cons 6. **Q. It would be interesting to see an experimental comparison against the version of LinTS which does not have covariance inflated by a factor of $d$.**
>
> In the current experiment for `LinTS`, we  tuned the value of $v$ which is multiplied to the covariance term and presented the best result over multiple candidate values for $v$.
> That is, we already covered the cases for `LinTS` where the covariance is inflated and not inflated through tuning and the best result is shown among the two.
> Hence, our experiments include the case where the reviewer suggested.
>
> [1] Hamsa Bastani, Mohsen Bayati, and Khashayar Khosravi. Mostly exploration-free algorithms for contextual bandits. Management Science, 67(3):1329–1349, 2021.

---

> ### Comment · Reviewer_Qbta · 2021-08-18
> **Updated review**
>
> I believe the authors have satisfactorily addressed several of my concerns. I have raised my score to a 7 and would recommend an accept for this paper.

---

### Official Review · Reviewer_Sp81 · 2021-07-16

**Rating:** 6
**Confidence:** 4

**Summary:**

This work studies the problem of contextual linear bandits following a combination of Thompson Sampling approach and Doubly Robust methods. It proposes an algorithm that improves the cumulative regret bound of existing LinTS algorithm by a factor of \sqrt{d} through the new definition of super-unsaturated arms.for some specific examples where the contexts are independently distributed.

**Limitations And Societal Impact:**

I believe this work does not have any potential negative societal impact.

**Main Review:**

The paper is well-written and the problem formulation and the contributions are clearly stated. My first comment is on the definition of \pi_i(t) which is crucial in computing Y_i^{DR}(t) and thus the algorithm implementation. The computation of \pi_i(t) is not discussed until in appendix A.2. I would suggest that the authors bring up this remark sometime earlier in the main body or make a small comment on it when they first introduce \pi_i(t) in line 144.

Questions:

1. Just to make sure, \tilde\pi_i(t) is P(\tilde Y_i(t)=\amax_{j\in[N]}\tilde Y_j(t))?
2. Could the authors comment on how they make sure that \pi_i(t) is non-zero as it is necessary in the computation of Y_i^{DR}(t)?
3. What part of the analysis does the following affect? If we consider that the action set is fixed at all rounds and replace the current definition of \phi^2 with \frac{1}{N}\sum_{i=1}^N X_i X_i^T.
4. Could the authors comment on the computation of imputation estimator and what the condition is to guarantee that \| \breve\beta_t-\beta\|\leq b for all t\in[T]? Can we simply replace \breve\beta_t with LSE of \beta, and b with O(\sqrt{d \log(T)})? In this case regret bound will become O(d^{3/2}\sqrt{T}). Do the authors have a better choice for \breve\beta_t and b?

**Time Spent Reviewing:**

4 hours

---

> ### Author Response · Authors · 2021-08-10
> **Responses to Reviewer Sp81**
>
> 1.**Q. Just to make sure, $\tilde\pi_i(t)$ is $P(\tilde Y_i(t)=\max_{j\in[N]}\tilde Y_j(t))$?**
>
> Yes.
>
> 2.**Q. Could the authors comment on how they make sure that $\pi_i(t)$ is non-zero as it is necessary in the computation of $Y_i^{DR}(t)$?**
>
> In lines 488-494 in supplementary materials, we prove that $\pi\_i(t)$ is nonzero.
> The proof is outlined by dividing into two cases: (i) when $\tilde{\pi}\_{i}(t)>\gamma$ for all $i=1,\ldots,N$, and (ii) when there exists $j$ such that $\tilde{\pi}\_j(t)\le \gamma$.
> On an abstract level, $\pi\_i(t)$ is nonzero, since $\tilde{\pi}\_i(t)$ is nonzero.
> Specifically, $\tilde{\pi}\_i(t)$ is nonzero because $\tilde{\pi}\_i(t)$ is computed as a tail probability of a multivariate normal distribution from which $\tilde{\beta}\_i(t)$ is sampled with support $\mathbb{R}^d$.
> Therefore, $\tilde{\pi}\_{i}(t):=\mathbb{P}[X\_{i}(t)^T\tilde{\beta}\_{i}(t)=\max\_{j} X\_{j}(t)^T\tilde{\beta}\_{i}(t) |\mathcal{H}\_t] > 0$ for any given contexts.
> Now we consider the case (i) such that $\tilde{\pi}\_{i}(t) > \gamma$ for all $i$.
> In case (i), we do not need resampling and thus $\pi\_{i}(t)=\tilde{\pi}\_{i}(t) > 0$.
> In case (ii), there are two types of arms: the first type is the arms such that $\tilde{\pi}\_i(t)\le\gamma$ and the second type is the arms such that $\tilde{\pi}\_i(t) > \gamma$.
> For the first type, arms with $\tilde{\pi}\_i(t) \le \gamma$, let $E$ be the event that resampling fails for $M\_t-1$ trials.
> Then
> $$
> \pi\_{i}(t) = \mathbb{P}( E, \\{\text{The } M\_t \text{-th candidate action is } i\\} ) = \tilde{\pi}\_{i}(t) \\{1-\sum\_{j:\tilde{\pi}\_{j}(t) > \gamma} \tilde{\pi}\_{j}(t)\\}^{M\_{t}-1}  > 0.
> $$
> The last inequality is due to $\sum\_{i:\tilde{\pi}_{i}(t) > \gamma}\tilde{\pi}\_{i}(t) < 1$, which holds since there exists an arm $i$ such that $\tilde{\pi}_{i}(t)\le \gamma$.
> For the second type, arms with $\tilde{\pi}_i(t)>\gamma$, $\pi_i(t)$ is derived in (15) which guarantees $\pi_i(t) > \tilde{\pi}_i(t)$ and thus, $\pi_i(t) > \gamma >0$.
> Thus the inverse probability $\pi_i(t)^{-1}$ is well-defined throughout our theoretical analysis.
>
> 3.**Q. What part of the analysis does the following affect? If we consider that the action set is fixed at all rounds and replace the current definition of $\phi^2$ with $\frac{1}{N}\sum_{i=1}^N X_i X_i^T$.**
>
> Thank you for bringing up the case of fixed contexts,  an important special case.
> Our results hold for this special case.
> The i.i.d. assumption and the assumption that $E(\sum_{i=1}^NX_i(\tau)X_i(\tau)^T)\ge N\phi^2$ are required to derive that $\lambda_{\text{min}}(V_t)\ge N\phi^2t$.
> With fixed contexts that satisfy $\lambda_{\text{min}}(\frac{1}{N}\sum_i{X_iX_i^T})\geq \phi^2$, the condition $\lambda_{\text{min}}(V_t)\ge N\phi^2t$ is straightforward, and consequently, Theorem 3 holds.
> We will add this in the revision.
>
> 4.**Q. Could the authors comment on the computation of imputation estimator and what the condition is to guarantee that $| \breve\beta_t-\beta|\leq b$ for all $t\in[T]$? Can we simply replace $\breve\beta_t$ with LSE of $\beta$, and $b$ with $O(\sqrt{d \log(T)})$? In this case regret bound will become $O(d^{3/2}\sqrt{T})$. Do the authors have a better choice for $\breve\beta_t$ and $b$?**
>
> In summary, choosing $\check{\beta}\_t$ as consistent estimators such as ridge estimator  used in our experiment or the LSE suggested by the reviewer render $||\check{\beta}\_t-\beta||\_2 = \tilde{O}(d/\sqrt{t})$.
> This is of a smaller order in $t$ than we assumed and its dependence on $d$ does not become an issue.
> In practice, it is natural to use consistent estimators such as a ridge estimator or the LSE that the reviewer suggested. In fact, in our experiment, we used $\check{\beta}\_t$ as a ridge estimator based on $\\{(X\_{a\_\tau}(\tau),Y\_{a\_{\tau}})\\}\_{\tau=1}^{t-1}$.
> If we replace $\check{\beta}\_t$ with the LSE of $\beta$, the order will be $||\check{\beta}\_t-\beta||\_2 = \tilde{O}(d/\sqrt{t})$.
> (We believe the reviewer was mistaken: $O(\sqrt{d\log t})$ is the bound of $||\check{\beta}\_t-\beta||\_{A\_t}$, where $A\_t=\sum\_{\tau=1}^{t}X\_{a\_\tau}(\tau)X\_{a\_\tau}(\tau)^T+\lambda I$.)
> This bound has a smaller order in $t$, and yields a term with an order that does not affect the overall rate.  Specifically, our regret depends on  $b$ through $C_{b,\sigma}$, where $C_{b,\sigma} = \frac{4+4b + 2(4b+2C\sigma)}{\phi^2}$.  Note that $b$ is linear in $C_{b,\sigma}$.
> When $\check{\beta}\_t$ is consistent,  $b=\tilde{O}(d/\sqrt{t})$ and then $C_{b,\sigma}$ can be partitioned into $C_{b,\sigma}= C_{\sigma}^* + \phi^{-2}\tilde{O}(d/\sqrt{t})$ where $C_{\sigma}^* = \frac{4 + 2(2C\sigma)}{\phi^2}$ is the main order term.
> Therefore, in this case, the dependence of $\check{\beta}\_t$ on $d$  does not affect the order of $C_{b,\sigma}$.
>
> Our condition, $||\check{\beta}_t-\beta||\_2 \le b$, covers worse cases than the two aforementioned examples and implies that imputation estimators are allowed to be biased.
> This tolerance to the biased imputation estimator is due to doubly-robust property and the fact $\pi_i(t)$ is known.
> Please see line 600 for the validity of doubly-robust estimating procedures when the imputation model is incorrectly specified and thus the imputation estimator is biased.
> If a biased estimator is used, we can rescale the estimator so that its $l_2$-norm is bounded by some constant $C>0$.
> Then, $||\check{\beta}_t-\beta||_2 \le ||\check{\beta}_t||_2 +||\beta||_2 \le C+1$ and $b = C+1 = O(1)$.

---

> > ### Comment · Reviewer_Sp81 · 2021-08-19
> > **Response**
> >
> > I thank the authors for their feedback. I have two more questions that I would be grateful if the authors could address.
> >
> > In your algorithm, the action $a_t$ is selected only when $\tilde \pi_{m_t}(t)>\gamma$ happens during the second loop. But, if this never happens during the $M_t$ number of tries, it is not specified how the action would be selected. Could the author comment on that?
> >
> > As to $C_{b,\sigma}$ and how $b$ can be chosen, as the authors mentioned, when $\breve\beta_t$ is a ridge estimator, then $||\beta-\breve\beta_t||\leq \tilde{\mathcal{O}}(d/\sqrt{t})$. However, due to the theorem statement, $b$ is a constant such that  $||\beta-\breve\beta_t||\leq b$ holds for all $t\in[T]$. Thus, $b$ must be chosen as $ \tilde{\mathcal{O}}(d)$. With this in mind, I am not convinced that  in $C_{b,\sigma}=C_{\sigma}^\ast+\phi^{-2}\tilde{\mathcal{O}}(d)$, where $C_{\sigma}^\ast = \frac{4+2(2C\sigma)}{\phi^2}$, $C_{\sigma}^\ast$ is the main order term. Could the authors please elaborate on this?

---

> > > ### Author Response · Authors · 2021-08-20
> > > **Additional response to Reviewer Sp81**
> > >
> > > **In your algorithm, the action $a\_t$ is selected only when $\tilde{\pi}\_{m\_t}(t) > \gamma$ happens during the second loop.
> > > But, if this never happens during the $M\_t$ number of tries, it is not specified how the action would be selected.
> > > Could the author comment on that?**
> > >
> > > When $\tilde{\pi}\_{m\_t}(t) > \gamma$ does not happens in $M\_t$ number of tries, the algorithm just pulls the $M\_t$-th candidate arm.
> > > But this case does not happen with high probability since we set $M\_t$ sufficiently large.
> > > (Please see line 452 in supplementary materials.)
> > > Since $\gamma < 1/N$ there exists at least one arm $j$ such that  $\tilde{\pi}\_j(t) > \gamma$.
> > > Thus, with sufficiently large $M\_t$, the algorithm can find $m\_t$ such that $\tilde{\pi}\_{m\_t}(t) > \gamma$.
> > > We will explicitly state this part in Algorithm 1 in the revision.
> > >
> > >
> > > **As to $C\_{b,\sigma}$ and how $b$ can be chosen, as the authors mentioned, when $\check{\beta} _t$ is a ridge estimator, then $||\beta - \check{\beta}\_t ||\_2 \le \tilde{O}(d/\sqrt{t})$.
> > > However, due to the theorem statement, $b$ is a constant such that $||\beta - \check{\beta}\_t ||\_2 \le b$ holds for all $t\in[T]$.
> > > Thus, $b$ must be chosen as $\tilde{O}(d)$.
> > > With this in mind, I am not convinced that in $C\_{b,\sigma} = C^{*}\_{\sigma} + \phi^{-2}\tilde{O}(d)$, where $C^{\ast}\_{\sigma}=\frac{4+2(2C\sigma)}{\phi^2}$, $C^{\ast}\_{\sigma}$ is the main order term.
> > > Could the authors please elaborate on this?**
> > >
> > > Thank you for raising this subtle point.
> > > In the initial response, we presented the bounds for $\check{\beta}\_t$ in two cases: (i) when  $\check{\beta}\_t$ is a consistent estimator and (ii) when $\check{\beta}\_t$ is a biased estimator, but this does not imply that the same case should be applied throughout $t\in[T]$.
> > > When $t$ is small, consistent estimators such as ridge estimators have nonnegligible bias, and the treatment can be that of the case (ii), not the case (i).
> > > Specifically, let $t\_d$ be the smallest integer that satisfies $\frac{\log t}{t} \le d^{-2}$, i.e. $t_d:=\inf\\{t\in[T]: \frac{\log t}{t} \le d^{-2} \\}$.
> > > For $t=1,\ldots,t\_d$, we can set $\check{\beta}\_t$ to be the rescaled ridge estimator so that its $l\_2$-norm is bounded by some constant $C>0$, i.e., $||\check{\beta}\_t||\_2 \le C$, and the resulting estimation error can be bounded using $C_{b,\sigma}$, with $b=O(1)$.
> > > For $t=t\_d+1,\ldots,T$, we set $\check{\beta}\_t$ as a standard ridge estimator with $||\check{\beta}\_t-\beta||\_2 \le O(d \sqrt{\log t / t} ) \le O(1)=b$ and the dominating term of the estimation error bound is $C^{\ast}\_{\sigma}$.

---

> > > > ### Comment · Reviewer_Sp81 · 2021-09-01
> > > > **One last question**
> > > >
> > > > Dear authors, thank you for your feedback. I have one last question about the imputation estimator $\check\beta_t$ that I would appreciate it if you could address. I am probably missing something, but it seems to me that the only requirement about $\check\beta_t$ is that $||\beta-\check\beta_t||_t\leq b$ and it is $H_t$-measurable, and as you mentioned it does not have to be an unbiased estimator. Why don’t you simply select $\check \beta_t$ as 0? In this case $b=||\beta||_2$. Am I missing something?

---

> > > > > ### Author Response · Authors · 2021-09-01
> > > > > **Response to the one last question**
> > > > >
> > > > > Selecting $\check{\beta}\_t$ as $0$ yields the regret bound stated in (3).
> > > > > Intuitively it is fine because the term $X\_i(t)^{T}\check{\beta}\_t$ in the definition of $Y^{DR}\_{i}(t)$ is multiplied with mean zero random variables, $\\{1-\frac{\mathbb{I}(i=a\_t)}{\pi\_i(t)}\\}$ and that is how we are protected against the bias of $\check{\beta}\_t$.
> > > > > As a result, the expectation of $Y^{DR}\_{i}(t)$ given $\mathcal{H}\_t$  remains unbiased as $X\_i(t)^T \beta$.
> > > > > This is the main strength of the doubly robust method in the bandit settings.
> > > > > For details, please see Section I.2 in supplementary materials.
> > > > >
> > > > > The main point of the condition on $\check{\beta}\_t$ is to demonstrate its robustness against choosing a biased $\check{\beta}\_t$.
> > > > > On the other hand, the quality of $\check{\beta}\_t$  influences the variance of the doubly robust estimator: when $\check{\beta}\_t$ is a consistent estimator, the doubly robust estimator becomes most efficient.
> > > > > Therefore, in practice, there is no reason to  use an inconsistent estimator intentionally.
> > > > > We recommend the ridge regression estimator for better empirical performances.
> > > > > We will add this point in the revision.

---

### Official Review · Reviewer_gHQb · 2021-07-16

**Rating:** 6
**Confidence:** 4

**Summary:**

The paper proposes a variant of Thompson Sampling for the linear contextual bandit problem. Using a doubly robust estimator for estimating the underlying parameter, the paper shows a regret bound of the order \sqrt{T}/\phi^2, where \phi^2 is a lower bound of the minimum eigenvalue of the covariance matrix of contexts. In some special cases, this bound implies a bound of the order d\sqrt{T}, which is an improvement of the current result.


**Limitations And Societal Impact:**

The authors have adequately addressed the limitations and potential negative societal impact of their work.

**Main Review:**

1. This paper presents an interesting application of the doubly robust estimator in the online setting (Thompson Sampling). Especially, it uses a resampling technique to provide the propensity score with a lower bound, without bringing auxiliary bias (when compared with other methods, say, propensity score clipping). The resulting bound improves the current result in some special cases.

2. How should one choose M_t when implementing the algorithm? How do the theoretical results and the empirical performance of the algorithm depend on the choice of M_t?

3. In line 534, is Lemma 4 applied conditional on the event {all \pi_{a_t} > \gamma}? In that case, does the conditions required by Lemma 4 still hold (e.g. sub-gaussian)?

3. Minor issue:
In Algorithm 1, when sampling another (\tilde{\beta}_1,...\tilde{\beta}_N) do we choose another m_t?



**Time Spent Reviewing:**

2 hours

---

> ### Author Response · Authors · 2021-08-10
> **Responses to Reviewer ghQb**
>
> 2.**Q. How should one choose $M_t$ when implementing the algorithm? How do the theoretical results and the empirical performance of the algorithm depend on the choice of $M_t$?**
>
> Thank you for highlighting the issue of selecting $M_t$.
> Discussion on $M_t$ can be found in the Section A.3 in supplementary materials.
> Below we outline how we select $M_t$.
> The main idea is that we choose $M_t$ to bound the probability that the resampling fails in finding an arm whose selection probability exceeds $\gamma$ in each round by $\delta/t^2$, i.e., $\mathbb{P}(a_t\notin\\{i:\tilde{\pi}_i(t) > \gamma\\})\le\delta/t^2$.
> We can bound the probability by choosing $M_t$ as a minimum integer that exceeds $\log\frac{t^2}{\delta}/\log\frac{1}{1-\gamma}$.
> Please see line 452 in supplementary materials.
> Intuitively, as $M_t$ increases, we have more opportunities for resampling and the probability that the resampling fails in finding an arm from $\\{i:\tilde{\pi}_i(t)>\gamma\\}$ decreases.
> Since $\gamma < 1/N$, there exists $j$ such that $\tilde{\pi}_j(t)>\gamma$, and the probability that the resampling fails is less than $1-\gamma$ in each resampling trial.
> Thus, by choosing $M_t$ for each round that satisfies $(1-\gamma)^{M_t}\le\delta/t^2$, the algorithm finds an arm $j$ such that $\tilde{\pi}_j(t)>\gamma$ in all rounds with high probability.
> We will add the empirical performance for various choices of $M_t$ in the revision.
>
> 3.**Q. In line 534, is Lemma 4 applied conditional on the event $\{{\rm all }~ \pi_{a_t} > \gamma\}$?
> In that case, does the conditions required by Lemma 4 still hold (e.g. sub-gaussian)?**
>
> We stated the joint probability not the conditional probability.  Specifically,
> denote $E(\tau):=\frac{\eta_{a_\tau}(\tau)}{\pi_{a_\tau}(\tau)}X_{a_\tau}(\tau)$ and $E^{(\gamma)}(\tau):=\frac{\mathbb{I}(\pi_{a_\tau}(\tau) > \gamma) \eta_{a_\tau}(\tau)}{\pi_{a_\tau}(\tau)}X_{a_\tau}(\tau)$.
> The event $G:=\cap_{t=1}^{T}\\{\pi_{a_t}(t)>\gamma\\}$ is not conditioned, but we derive a joint probability as follows: for any $x>0$, we have
> $$
> \mathbb{P}(||\sum\_{\tau=1}^{t} E(\tau)\||\_2 > x, G ) \le \mathbb{P} (||\sum\_{\tau=1}^{t}E^{(\gamma)}(\tau) )||\_2 > x, G ) \le
> \mathbb{P} (||\sum\_{\tau=1}^{t}E^{(\gamma)}(\tau) )||\_2 > x).
> $$
> By Assumption 2, the distribution of $\eta\_i(\tau)$ given $\mathcal{H}\_{\tau}$ is independent of $a\_{\tau}$.
> Thus the random variable $\frac{\mathbb{I}(\pi_{a_\tau}(\tau) > \gamma) \eta_{a_\tau}(\tau)}{\pi_{a_\tau}(\tau)}$ is sub-Gaussian given $\mathcal{H}\_\tau$ and $a\_{\tau}$.
>
> 4.**Minor issue: In Algorithm 1, when sampling another $(\tilde{\beta}_1,...\tilde{\beta}_N)$ do we choose another $m_t$?**
>
> Yes. We will explicitly state  this in Algorithm 1 in the revision.

---

> > ### Comment · Reviewer_gHQb · 2021-09-01
> > **Response to the author**
> >
> > I would like to thank the authors for their response. My concerns have been addressed, and I would like to raise my score to 7.

---

### Author Response · Authors · 2021-08-10
**Response for All Reviewers**

We thank the reviewers for taking time to read our paper and appreciate constructive and helpful comments.
In this **response for all reviewers**, we highlight the issues commented by more than one reviewer.

(i) Response regarding the number of maximum possible resampling $M _t$.

We denote by $M_t$ the number of maximum possible resampling. Two reviewers queried about how to select $M_t$.
Section A.3 in supplementary materials describes issues related to $M_t$ including a suitable choice of  $M_t$.
Below we outline how we select $M_t$.

Our algorithm attempts resampling up to $M_t$ times  to find an arm in $\\{i:\tilde{\pi}_i(t)>\gamma\\}$.
The main point in selecting $M_t$ is to bound the probability that the resampling fails in finding an arm whose selection probability exceeds $\gamma$ for some $\delta$, i.e., $\mathbb{P}(a_t\notin\\{i:\tilde{\pi}_i(t) > \gamma\\})\le\delta/t^2$.
We can bound this probability by choosing $M_t$ as a minimum integer that exceeds $\log\frac{t^2}{\delta}/\log\frac{1}{1-\gamma}$.
For detailed derivations, please see line 452 in supplementary materials.
Intuitively, as $M_t$ increases, we have more opportunities for resampling and the probability that the resampling fails in finding arms in $\\{i:\tilde{\pi}_i(t)>\gamma\\}$ decreases.
Since $\gamma < 1/N$, there exists $j$ such that $\tilde{\pi}_j(t)>\gamma$, and the probability that the resampling fails is less than $1-\gamma$ in each resampling trial.
Thus, by choosing $M_t$ for each round that satisfies $(1-\gamma)^{M_t}\le\delta/t^2$, the algorithm finds an arm $j$ such that $\tilde{\pi}_j(t) > \gamma$ in all rounds with high probability.

Selecting an arm from the set $\\{i:\tilde{\pi}_i(t) > \gamma\\}$ with high probability is crucial in achieving the regret bound of order $\tilde{O}(\phi^{-2}\sqrt{T})$  for two reasons.
First, due to the relationship $\\{i:\tilde{\pi}_i(t) > \gamma\\} \subseteq \\{i:\text{The arm } i\text{ is super-unsaturated}\\}$, it guarantees that the arm is super-unsaturated and our novel regret decomposition (6) holds to achieve our regret bound.
Second, since $\pi _{a_t}(t) \ge \tilde{\pi} _{a_t}(t)$ holds when $a_t \in \\{i:\tilde{\pi} _i(t) > \gamma\\}$, the inverse probability, $\pi _{a_t}(t)^{-1}$ is also bounded by $\gamma^{-1}$ which appears in $Y^{DR}_i(t)$ and the proof of Theorem 3.

(ii) Response regarding the imputation estimator $\check{\beta}_t$.

About the imputation estimator $\check{\beta}_t$, we assume that $||\check{\beta}_t-\beta||_2 \le b$, where $b=O(1)$.
Questions were raised by two reviewers whether this condition is achievable and what estimators guarantee this condition, and how it affects the regret bound.
First, consistent estimators such as ridge estimator used in our experiments or the least squared estimator suggested by one of the reviewers satisfy our assumption since $||\check{\beta}_t-\beta||_2=\tilde{O}(d/\sqrt{t})$.
These estimators yield a smaller order in $t$ than we assumed  and its dependence on $d$ does not become an issue.
Specifically, our regret depends on  $b$ through $C _{b,\sigma}$, where $C _{b,\sigma} = \frac{4+4b + 2(4b+2C\sigma)}{\phi^2}$.
Note that $b$ is linear in $C _{b,\sigma}$.
When $\check{\beta} _t$ is consistent, $b=\tilde{O}(d/\sqrt{t})$ and then $C _{b,\sigma}$ can be partitioned into $C _{b,\sigma}= C _{\sigma}^* + \phi^{-2}\tilde{O}(d/\sqrt{t})$ where $C _{\sigma}^* = \frac{4 + 2(2C\sigma)}{\phi^2}$ is the main order term.
Therefore the dependence of $\check{\beta} _t$ on $d$ does not affect the order of $C _{b,\sigma}$.

Our condition, $||\check{\beta} _t-\beta|| _2 \le b$,  covers worse cases than the case of consistent estimators and implies that the imputation estimators are allowed to be  biased.
This tolerance to the biased imputation estimator is due to doubly-robust property and the fact $\pi _i(t)$ is known.
Please see line 600 for the validity of doubly-robust estimating procedures when the imputation model is incorrectly specified and thus the imputation estimator is biased.
If a biased estimator is used, we can rescale the estimator so that its $l_2$-norm is bounded by some constant $C>0$.
Then, $||\check{\beta}_t-\beta|| _2 \le ||\check{\beta}_t||_2 +||\beta||_2 \le C+1$ and $b = C+1 = O(1)$.
We will reflect this in the revision.

---

### Decision · Program_Chairs · 2021-09-27

**Decision:**

Accept (Spotlight)

**Comment:**

This paper combines Thompson Sampling and doubly robust estimators for the linear contextual bandits. We have had many discussions and comments. The authors successfully answered all the questions raised during the rebuttal.  All the reviewers agree that this is a nice paper with a solid contribution.